# LESS IS MORE: LEARNING REFERENCE KNOWLEDGE USING NO-REFERENCE IMAGE QUALITY ASSESSMENT

## ABSTRACT

*Image Quality Assessment* (IQA) with reference images has achieved great success by imitating the human vision system, in which the image quality is effectively assessed by comparing the query image with its pristine reference image. However, for the images in the wild, it is quite difficult to access accurate reference images. We argue that it is possible to learn reference knowledge under the *No-Reference Image Quality Assessment* (NR-IQA) setting, which is effective and efficient empirically. Concretely, by innovatively introducing a novel feature distillation method in IQA, we propose a new framework to learn comparative knowledge from non-aligned reference images. Then, we further propose inductive bias regularization to inject different inductive biases into the model to achieve fast convergence and avoid overfitting. Such a framework not only solves the congenital defects of NR-IQA but also improves the feature extraction framework, enabling it to express more abundant quality information. Surprisingly, our method utilizes less input—eliminating the need for reference images during inference—while obtaining more performance compared to some IQA methods that do require reference images. Comprehensive experiments on eight standard IQA datasets show that our approach outperforms state-of-the-art NR-IQA methods.

## 1 INTRODUCTION

*Image Quality Assessment* (IQA)(Saad et al., 2012; Mittal et al., 2012; Zhang et al., 2015) has been extensively applied in various computer vision tasks, including image restoration(Banham & Katsaggelos, 1997) and super-resolution (Dong et al., 2015). By mimicking the human vision system (HVS), IQA methods effectively estimate the quality of a query image using its pristine reference image, yielding promising results with appropriate data support. For instance, a seminal study (Wang et al., 2004) introduced a structural similarity index method that utilizes all or part of the information from *High Quality* (HQ) reference images to evaluate image quality, marking substantial progress toward establishing Full-Reference Image Quality Assessment (FR-IQA). Learning-based FR-IQA methods such as IQT (Cheon et al., 2021) further improve accuracy by comparing pixel-aligned HQ reference images with distorted ones. However, pristine reference images are rarely available in real-world scenarios, limiting the practical application of FR-IQA.

To address the challenge of lacking reference images, No-Reference Image Quality Assessment (NR-IQA) techniques (Zhang et al., 2023b; 2021) have been developed to evaluate image quality solely based on the input image. Recently, the growing success of Vision Transformers (ViT)(Dosovitskiy et al., 2021) has led to state-of-the-art NR-IQA methods(Golestaneh et al., 2022; Ke et al., 2021; Qin et al., 2023) adopting ViT-based architectures, optimizing feature extraction and quality regression in an end-to-end manner. Although this resolves the issue of missing reference images, the performance of these methods remains suboptimal. Psychological studies have demonstrated that the HVS more effectively perceives image quality by comparing multiple images rather than assessing a single image in isolation (Sheikh & Bovik, 2006). As a result, NR-IQA methods that do not leverage comparative information between HQ and low-quality (LQ) images tend to underperform (Yin et al., 2022).

Another line of work has attempted to reduce the dependency on reference images while retaining compatibility with the HVS's comparative mechanisms. For instance, methods like those in (Liang et al., 2016) introduced non-aligned reference images, which relax the need for pixel-perfect alignment but still require content similarity. Later work (Yin et al., 2022) extended this idea to allow reference

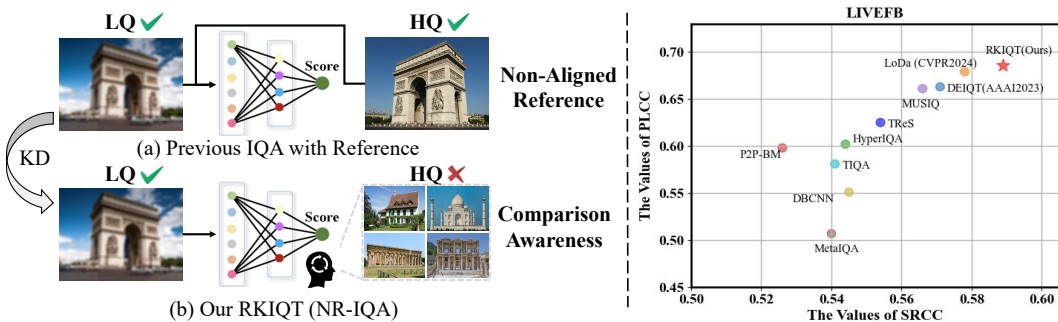

Figure 1: The proposed RKIQT outperforms all existing NR-IQA methods because it develops an awareness of quality comparison with high-quality images during the knowledge distillation process. Note that it also exceeds some IQA methods that do require reference images, as shown in Tab. 2.

images without content similarity. However, it still require finding suitable HQ images as references during inference, adding computational complexity and limiting scalability. This motivates our work:

*Can NR-IQA benefit from the comparison knowledge used in reference-based IQA methods, while eliminating the need for reference images during inference?*

Furthermore, the popular ViT model excels at modeling non-local dependencies (Qin et al., 2023; Golestaneh et al., 2022), but it demonstrates weakness in handling local structures and inductive biases (Yuan et al., 2021; Cordonnier et al., 2019). This limits the potential of the ViT for the NR-IQA task, which heavily relies on local and non-local features (Su et al., 2020) and often lacks large amounts of available training data (Zhang et al., 2023b; Zhao et al., 2023b). Consequently, previous works (Golestaneh et al., 2022; Xu et al., 2024) have shown the benefits of using local features extracted from convolutional neural network (CNN) networks for enhancing ViT. However, integrating different model architectures inevitably increases the inference cost and the risk of overfitting (Raghu et al., 2021; Naseer et al., 2021), particularly in IQA tasks with small datasets.

In this paper, we propose a novel NR-IQA framework called the *Reference Knowledge-Guided Image Quality Transformer* (RKIQT). This framework leverages reference information and rich inductive biases acquired during knowledge distillation to perform IQA inference without the need for high-quality reference images, as shown in Fig. 1. To comprehensively understand the differences between high-quality and low-quality images and to develop a comparative awareness, we introduce a novel *Masked Quality-Contrastive Distillation* (MCD) method. This method guides the student model to emulate the teacher's prior comparison information based on partial feature pixels. Furthermore, to adjust the inductive biases of the ViT, ensuring rapid convergence and preventing overfitting, we propose an inductive bias regularization method. This technique adds two learnable tokens to the ViT encoder and employs a reverse distillation strategy to learn beneficial knowledge from both a CNN teacher and an Involution Neural Network (INN) (Li et al., 2021) teacher. It integrates complementary inductive biases from convolution (spatial-agnostic and channel-specific) and involution (spatial-specific and channel-agnostic) into the ViT, thereby enriching its representation with local and global quality-aware features. After training the student, it can predict the quality of test images without requiring any reference images. Our contributions are summarized as follows:

- We creatively use feature distillation in the NR-IQA setting to achieve comparative knowledge. This method requires ***less*** input by eliminating the need for reference images during inference, yet it achieves a ***more*** impressive performance compared to some traditional IQA methods that rely on reference images.

- For feature distillation, we introduce a Masked Quality-Contrastive Distillation method to guide the student model in emulating the teacher's prior comparison information based on partial feature pixels, resulting in a more robust model with stronger representation capacity.

- For regularization, we leverage the reverse distillation strategy while distilling teachers and tokens with different inductive biases while speeding up the training process, we adapt students to this reverse distillation to obtain more competitive quality-aware benefits by fine-tuning the quality-aware ability of pre-trained teachers.

## 2 RELATED WORK

**NR-IQA with Deep Learning**. The deep learning methods have achieved extraordinary success in various computer vision tasks, which by nature attracts a great deal of interest in utilizing deep learning for IQA tasks. The early version of deep learning-based IQA method (Zhang et al., 2018b; Su et al., 2020) is based on the convolutional neural network (CNN) (He et al., 2016) thanks to its powerful feature expression ability. The CNN-based IQA method generally treats the IQA task as the downstream task of object recognition, following the standard pipeline of pre-training and fine-tuning. Such a strategy is useful as these pre-trained features share a certain degree of similarity with the quality-aware features of images (Su et al., 2020). Recently, the Vision Transformer (ViT) (Dosovitskiy et al., 2021) based NR-IQA methods are growing in popularity, owing to the strong capability of ViT in modeling the non-local perceptual features of the image. There are mainly two types of architectures for the ViT-based NR-IQA methods, including hybrid Transformer (Golestaneh et al., 2022; Xu et al., 2024) and pure ViT-based Transformer (Ke et al., 2021). The hybrid architecture generally combines the CNNs with the Transformer, which are responsible for the local and long-range feature characterization, respectively. The ViT-based methods can be further exploited. Nevertheless, transformers have fewer inductive biases than CNNs (e.g., translation equivariance and locality) and thus suffer when the given amounts of training data are insufficient (Dosovitskiy et al., 2021).

**Knowledge Distillation.** Recent advancements in knowledge distillation have been significant. (Hinton et al., 2015) laid the foundational concept of training a smaller "student" model to replicate a larger "teacher" model. (Mirzadeh et al., 2020) added the concept of an assistant network that aims to narrow the gap between teachers and students, thus improving the effectiveness of distillation. Recent works in IQA, such as (Yue et al., 2022), have utilized mutual learning to improve IQA performance in small sample scenarios. (Zheng et al., 2021) and (Yin et al., 2022) have explored using KD to transfer reference information to student models. This approach aims to reduce the student models' dependency on the availability of reference images, leading to the development of degraded-reference IQA (DR-IQA) and non-aligned reference IQA (NAR-IQA) methods. However, these methods still face limitations due to their reliance on reference images, which is impractical for NR-IQA. To the best of our knowledge, we make the first attempt to transfer more HQ-LQ difference prior information and rich inductive biases to the NR-IQA via KD, endowing students with the awareness of comparison. Experiments prove that distillation operations can further help our students achieve more accurate and stable performance.

## 3 METHODOLOGY

To clarify, we use bold formatting to denote vectors (e.g., $\boldsymbol{x}, \boldsymbol{y}$), matrices (e.g., $\boldsymbol{X}, \boldsymbol{Y}$), or tensors. Additionally, we define some common notations in image quality assessment (IQA). In particular, we define the Low Quality (LQ) image to be estimated as $I_L$, the randomly selected annotated High-Quality (HQ) image as $I_H$, the feature map of the network output as $\boldsymbol{F}$, the quality prediction of network $\boldsymbol{N}$ is denoted as $Y$.

IQA is highly correlated to subjective cognition, which is more accurate when the pristine reference image is provided (Wang et al., 2004). However, it is impractical to find reference images in real-world applications. In this paper, we propose a novel framework that learns reference information under the NR-IQA setting. It consists of three dedicatedly designed components: (i) The NR-student Reference Knowledge-guided Image Quality Transformer (RKIQT) $\boldsymbol{N}_s$ is the main network of our method, which receives the knowledge from other teacher networks. (ii) The non-aligned reference teacher (NAR-teacher) $\boldsymbol{N}_{T_{nar}}$ offers the comparison knowledge to $\boldsymbol{N}_s$ by Masked Quality-Contrastive Distillation. (iii) The inductive bias teachers $\boldsymbol{N}_{T_{conv}}, \boldsymbol{N}_{T_{inv}}$ provide the inductive bias from convolution and involution (Li et al., 2021) knowledge to $\boldsymbol{N}_s$ by Inductive Bias Regularization.

As illustrated in Fig. 2, given input images, our student and NAR-teacher first obtain the LQ local-global fused features and the HQ-LQ distribution difference through the outputs of the transformer encoder, respectively. The student's feature map is first masked and then used to reconstruct a new feature through a simple generation module, which is supervised by the teacher (Sec. 3.2). Then, we further propose inductive bias regularization, which extracts local and global knowledge from CNN and Involution (Li et al., 2021) respectively to achieve fast convergence and avoid overfitting (Sec. 3.3). After training, all teacher distillation and regularization will be deprecated, the student model is capable of directly assessing the quality of the input images without reference.

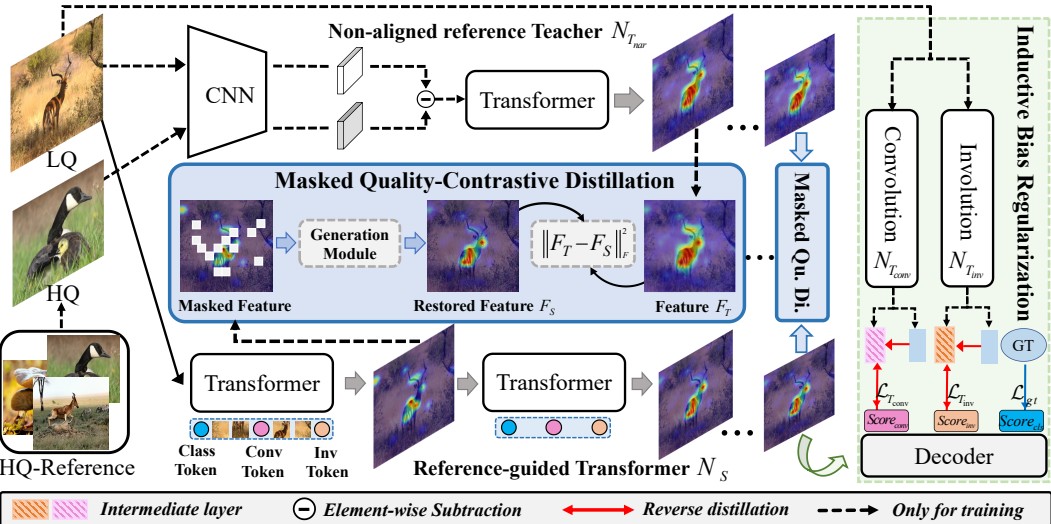

Figure 2: The overview of our RKIQT. We first mask the feature map of the student network, which is then used to generate the new feature that is supervised by a non-aligned reference teacher network (Sec. 3.2). After that, we further propose inductive bias regularization, which extracts local and global knowledge from CNN and involution to achieve fast convergence and avoid overfitting (Sec. 3.3).

## 3.1 STUDENT AND TEACHER ARCHITECTURE DESIGN

**Non-aligned reference Teacher.** Inspired by previous work (Yin et al., 2022), we utilize a non-aligned reference IQA teacher (NAR-teacher) to provide reliable comparison knowledge during training, as it only needs high-quality images with arbitrary content as reference images and no reference images with specific pixel alignments. This also further narrows the training cost of our method. Our NAR-teacher network employs a pre-trained Inception-ResNet-v2 (Szegedy et al., 2017) to extract feature maps from both unaligned reference and distorted input images. It then computes the difference features between these two sets of features and transforms them into a 1D patch sequence. This sequence serves as the input to a ViT (Dosovitskiy et al., 2021) encoder, which constructs globally aware difference features. By comparing the unaligned reference image with the input image, the NAR-teacher network provides valuable comparative knowledge through offline knowledge distillation, optimizing the student network for NR-IQA tasks.

**Reference-guided Transformer Student**. As mentioned before, we propose cross-inductive bias teachers that can focus on various inductive biases (Sec. 3.3) to achieve fast convergence and prevent overfitting. To align additional learnable tokens with different inductive bias teachers, we introduce token inductive bias alignment. We use three tokens: Class token, Conv token, and Inv token. To eliminate the inductive bias in the Class token, we apply truncated Gaussian initialization, which ensures values are drawn from a neutral, unbiased distribution. On the other hand, we introduce the corresponding inductive bias into the remaining two tokens. The Conv token and Inv token use the average pooling outputs of convolution stem and involution stem, respectively, with added position embeddings. The output of the encoder includes three inductive bias tokens denoted by $\hat{F}_o \in \mathbb{R}^{3 \times D}$. Then, we follow previous work (Qin et al., 2023) by introducing a transformer decoder to further decode inductive biases Class, Conv, and Inv tokens through multi-head self-attention (MHSA), thus making the extracted features more significant and comprehensive to the image quality. Finally, the outputs of the Class token, Conv token, and Inv token are supervised by the ground truth and corresponding inductive bias teacher. For further details, refer to Sec. A.4.

## 3.2 MASKED QUALITY-CONTRASTIVE DISTILLATION

We make the first attempt to transfer HQ-LQ differential prior information from non-aligned reference teacher (Sec. 3.1) to NR-IQA via Knowledge Distillation (KD). Traditional KD methods require the student model to directly mimic the teacher model's output. Such a mechanism is not suitable for our method, since our student model lacks reference images, it can only mine the quality features

of LQ images. It is misaligned with the HQ-LQ distribution difference features captured by the teacher. Direct imitation teacher's output may introduce negative regularization that degrades the final performance and stability (Li et al., 2023) (refer to Tab. 4). Empirically, we have identified that these negative effects stem mainly from two aspects: (1) the traditional mean squared error (MSE) loss directly aligns features on a one-to-one basis, increasing the training difficulty (Li et al., 2023); (2) The quality differences feature between high-quality (HQ) and low-quality (LQ) images tend to appear in salient regions (Varga, 2022). This means that non-salient pixel features often miss the chance to learn from reference knowledge, limiting the model's ability to generalize to various types of distorted images.

Inspired by the masking mechanisms (He et al., 2022; Yang et al., 2022), this paper proposes a simple yet effective feature distillation method, named Masked Quality Contrastive Distillation (MCD). The goal of the MCD is not to directly mimic the HQ-LQ difference features extracted by the teacher but rather to use these features to guide the student in developing a comparative awareness. Specifically, we first randomly mask the student features and then force the student model to generate the teacher's complete features based on partial pixels through a simple feature generation module. Benefiting from the MCD module, the enhancement of the student's comparative awareness is reflected in two key aspects. First, reconstructing teacher features from masked segments rather than direct imitation not only improves the student model's ability to perceive local image contrast (He et al., 2022) but also reduces training difficulty. Second, in each iteration, the MCD method randomly masks portions of the feature map's pixels. This ensures that all pixels are used throughout the training process to learn reference knowledge.

Specifically, for a given $i$-th image, all layer features $\boldsymbol{F}_T^{(i)}$ from the NAR-teacher are utilized to guide the training of the NR-student. Initially, we define a random mask function $M(\cdot)$ to obscure the corresponding features of the student that have been processed through an adaptive layer, aligning them with the teacher's feature map. Subsequently, the student's features are used to generate new feature maps via a generation module $\mathcal{G}(\cdot)$, which comprises two 3×3 convolutional layers with ReLU activation functions. Finally, mean squared error (MSE) loss is employed as the feature distillation loss to transfer knowledge to the corresponding layer features $\boldsymbol{F}_S^{(i)}$ of the NR-student. This process can be expressed as follows:

$$\boldsymbol{F}_S^{(i)} = \mathcal{G}(M(\boldsymbol{F}_{S'}^{(i)}))$$

$$\mathcal{L}_{\text{feature}}(\boldsymbol{F}_S, \boldsymbol{F}_T) = \frac{1}{K} \sum_{i=1}^{K} \|\boldsymbol{F}_T^{(i)} - \boldsymbol{F}_S^{(i)}\|_F^2 \tag{1}$$

where $\boldsymbol{F}_{S'}^{(i)}$ represent the aligned feature map of the student encoder , $K$ denotes the number of images in the training set. Guided by MCD, our student effectively learns more HQ-LQ difference knowledge and remains stable across differently distorted images.

### 3.3 Inductive Bias Regularization

Prior research (Dosovitskiy et al., 2021) found that transformers have fewer inductive biases and thus suffer when the given amounts of training data are insufficient. This issue can be addressed through the logits distillation technique (Zhu et al., 2018), where a student model with smaller inductive biases can learn various knowledge from teachers with different inductive biases (Touvron et al., 2021). Therefore, to achieve fast convergence and avoid overfitting, we propose an inductive bias regularization that adopts EfficientNet-b0 (Tan & Le, 2019) and RedNet101 (Li et al., 2021) (pre-trained on ImageNet (Deng et al., 2009)) which considers the trade-off between accuracy and complexity to guide the student's logits output to obtain more comprehensive representation power. [1]. To explain, CNN has a strong locality modeling capability, while the involution kernel is shared across channels but distinct in the spatial extent, and dynamically generating kernel parameters, which enables the extraction of long-range spatial information in images. In this way, the knowledge from teachers compensates for each other and significantly improves the accuracy of our RKIQT.

However, we believe that if teachers' logits with different inductive biases are directly used to supervise students, there will be a relatively large quality perception gap between teacher and

---

[1]We do not use ViT as a teacher to guide long-range information because it has fewer inductive biases (Ren et al., 2022)

student (refer to Table 7). Therefore, we introduce a learnable intermediate layer to solve such a problem. Specifically, the introduced learnable intermediate layer is proposed to aggregate the output of the corresponding teacher network and also takes the supervision information from the student network. Take the INN branch as an example (same with CNN branches), given the $i$-th image, the teacher's output is defined as $\boldsymbol{Y}_{T'_{inv}}$. Meanwhile, the output of the teacher's learnable intermediate layer and student network is defined as $\boldsymbol{Y}_{T_{inv}}$ and $\boldsymbol{Y}_{S_{inv}}$, respectively, which is expressed as follows:

$$\boldsymbol{Y}_{T_{inv}} = \text{MLP}((A_1(F_1) \oplus A_2(F_2)) \oplus A_3(F_3)) \tag{2}$$

$(F_1, F_2, F_3)$ are features from different middle layers of the pre-trained Teacher network, transformed by the adaptation layer $\boldsymbol{A}(\cdot)$ and feature addition $\oplus$. During training, $\mathcal{L}_1$ regression is used as the distillation loss, and the loss function for the student and intermediate layers is:

$$\mathcal{L}_{S_{inv}} = \frac{1}{K} \sum_{i=1}^{K} \|\boldsymbol{Y}_{S_{inv}}^{(i)} - \boldsymbol{Y}_{T_{inv}}^{(i)}\|_1 \tag{3}$$

$$\mathcal{L}_{T_{inv}} = \frac{1}{K} \sum_{i=1}^{K} \|\boldsymbol{Y}_{T_{inv}}^{(i)} - \boldsymbol{Y}_{T'_{inv}}^{(i)}\|_1 + \frac{1}{K} \sum_{i=1}^{K} \|\boldsymbol{Y}_{T_{inv}}^{(i)} - \boldsymbol{Y}_{S_{inv}}^{(i)}\|_1 \tag{4}$$

where $\mathcal{L}_{S_{inv}}$ and $\mathcal{L}_{T_{inv}}$ donates the supervision loss of the student and teacher's intermediate layer. In this way, the ability gap between teachers and students is effectively narrowed. Meanwhile, the students even outperform the teacher and get a noticeable improvement. From the perspective of a student, the output takes supervision from two teachers, which is formally defined as:

$$\mathcal{L}_{\text{Logits}} = \mathcal{L}_{S_{inv}} + \mathcal{L}_{S_{conv}}, \tag{5}$$

where the calculation process of $\mathcal{L}_{S_{conv}}$ is similar to $\mathcal{L}_{S_{inv}}$. Take the ground truth as extra supervision, the loss function of the student () is finally formally defined as:

$$\mathcal{L} = \frac{1}{K} \sum_{i=1}^{K} \|\boldsymbol{Y}_{gt}^{(i)} - \boldsymbol{N_s}(\boldsymbol{I}_L^{(i)})\|_1 + \lambda_1 \mathcal{L}_{\text{Feature}} + \lambda_2 \mathcal{L}_{\text{Logits}}, \tag{6}$$

where $\boldsymbol{I}_L^{(i)}$ is the $i_{\text{th}}$ distorted image, $N_s(\cdot)$ is the student predicted results and labeled ground-truth is represented as $\boldsymbol{Y}_{gt}^{(i)}$. $\lambda_1, \lambda_2$ are the hyperparameters.

## 4 EXPERIMENTS

### 4.1 DATASETS AND EVALUATION CRITERIA

We evaluate the proposed RKIQT on eight typical datasets, including four synthetic datasets (LIVE (Sheikh et al., 2006), CSIQ (Larson & Chandler, 2010), TID2013 (Ponomarenko et al., 2015), KADID (Lin et al., 2019)) and four authentic datasets (LIVEC (Ghadiyaram & Bovik, 2015), KonIQ (Hosu et al., 2020), LIVEFB (Ying et al., 2020), SPAQ (Fang et al., 2020)). Authentic datasets contain diverse real-world images, while synthetic datasets feature distorted images with various degradation types. Performance is measured using Spearman's Rank Correlation Coefficient (SRCC) and Pearson's Linear Correlation Coefficient (PLCC). SRCC and PLCC values range from -1 to 1, with superior performance indicated by absolute values close to one.

### 4.2 IMPLEMENTATION DETAILS

We build the Transformer encoder based on ViT-S from DeiT III (Touvron et al., 2022), with an encoder depth of 12 and 6 heads. The decoder depth is set to 1. The model is trained for 9 epochs with a learning rate of $2 \times 10^{-4}$. For each dataset, 80% of the images are for training and 20% for testing, repeating this 10 times to mitigate bias and report the average SRCC and PLCC. In addition, during training, randomly sample high-quality images from the DIV2K HR dataset (Agustsson & Timofte, 2017) as reference inputs for the NAR-teacher network. These experiments were performed on four NVIDIA 3090 GPUs. During training on any of the 8 IQA datasets, the student network should use the corresponding pre-trained CNN and INN teachers for that dataset, while the NAR-teacher is pre-trained exclusively on the synthetic KADID dataset. The teacher performs offline distillation during student training. Please refer to appendix A.3 for more details.

Table 1: Performance comparison measured by averages of SRCC and PLCC, compared with NR-IQA. Bold entries indicate the best results, underlines indicate the second-best.

| Method (*Infer Params (M)*) | LIVE | | CSIQ | | TID2013 | | KADID | | LIVEC | | KonIQ | | LIVEFB | | SPAQ | |
|---|---|---|---|---|---|---|---|---|---|---|---|---|---|---|---|---|
| | PLCC | SRCC | PLCC | SRCC | PLCC | SRCC | PLCC | SRCC | PLCC | SRCC | PLCC | SRCC | PLCC | SRCC | PLCC | SRCC |
| DBCNN (Zhang et al., 2018b) | 0.971 | 0.968 | 0.959 | 0.946 | 0.865 | 0.816 | 0.856 | 0.851 | 0.869 | 0.851 | 0.884 | 0.875 | 0.551 | 0.545 | 0.915 | 0.911 |
| TIQA (You & Korhonen, 2021) | 0.965 | 0.949 | 0.838 | 0.825 | 0.858 | 0.846 | 0.855 | 0.85 | 0.861 | 0.845 | 0.903 | 0.892 | 0.581 | 0.541 | - | - |
| MetaIQA (Zhu et al., 2020) | 0.959 | 0.960 | 0.908 | 0.899 | 0.868 | 0.856 | 0.775 | 0.762 | 0.802 | 0.835 | 0.856 | 0.887 | 0.507 | 0.54 | - | - |
| HyperIQA (Su et al., 2020) | 0.966 | 0.962 | 0.942 | 0.923 | 0.858 | 0.840 | 0.845 | 0.852 | 0.882 | 0.859 | 0.917 | 0.906 | 0.602 | 0.544 | 0.915 | 0.911 |
| TReS (*152M*) (Golestaneh et al., 2022) | 0.968 | 0.969 | 0.942 | 0.922 | 0.883 | 0.863 | 0.858 | 0.859 | 0.877 | 0.846 | 0.928 | 0.915 | 0.625 | 0.554 | - | - |
| MUSIQ (*27M*) (Ke et al., 2021) | 0.911 | 0.940 | 0.893 | 0.871 | 0.815 | 0.773 | 0.872 | 0.875 | 0.746 | 0.702 | 0.928 | 0.916 | 0.661 | 0.566 | 0.921 | 0.918 |
| Re-IQA (*48M*) (Saha et al., 2023) | 0.971 | 0.970 | 0.960 | 0.947 | 0.861 | 0.804 | 0.885 | 0.872 | 0.854 | 0.840 | 0.923 | 0.914 | - | - | 0.925 | 0.918 |
| DEIQT (*24M*) (Qin et al., 2023) | 0.982 | 0.980 | 0.963 | 0.946 | 0.908 | 0.892 | 0.887 | 0.889 | 0.894 | 0.875 | 0.934 | 0.921 | 0.663 | 0.571 | 0.923 | 0.919 |
| LIQE (*151M*) (Zhang et al., 2023b) | 0.951 | 0.970 | 0.939 | 0.936 | - | - | 0.931 | 0.930 | 0.910 | 0.904 | 0.908 | 0.919 | - | - | - | - |
| LoDa (*120M*) (Xu et al., 2024) | 0.979 | 0.975 | - | - | 0.901 | 0.869 | 0.920 | 0.912 | 0.899 | 0.876 | 0.933 | 0.920 | 0.679 | 0.578 | 0.928 | 0.925 |
| RKIQT (*28M*) | 0.986 | 0.984 | 0.970 | 0.958 | 0.917 | 0.900 | 0.911 | 0.911 | 0.917 | 0.897 | 0.943 | 0.929 | 0.686 | 0.589 | 0.928 | 0.923 |

Table 2: Model comparisons on standard IQA datasets trained on the synthetic Kaddid-10K dataset, with FR-IQA and NAR-IQA results reported from a previous study (Yin et al., 2022).

| IQA Type | Method | LIVE | | | CSIQ | | | TID2013 | | | KonIQ-10K | | |
|---|---|---|---|---|---|---|---|---|---|---|---|---|---|
| | | SRCC | PLCC | KRCC | SRCC | PLCC | KRCC | SRCC | PLCC | KRCC | SRCC | PLCC | KRCC |
| FR-IQA | LPIPS (Zhang et al., 2018a) | 0.932 | 0.934 | 0.765 | 0.876 | 0.896 | 0.689 | 0.670 | 0.749 | 0.497 | - | - | - |
| | DISTS (Ding et al., 2022) | 0.954 | 0.954 | 0.811 | 0.929 | 0.928 | 0.767 | 0.830 | 0.855 | 0.639 | - | - | - |
| | IQT (Cheon et al., 2021) | 0.970 | - | 0.849 | 0.943 | - | - | 0.899 | - | 0.717 | - | - | - |
| NAR-IQA | DCNN (Liang et al., 2016) | 0.752 | 0.756 | 0.594 | 0.721 | 0.716 | 0.563 | 0.473 | 0.492 | 0.346 | 0.258 | 0.256 | 0.147 |
| | WaDIQaM (Bosse et al., 2017) | 0.897 | 0.894 | 0.707 | 0.799 | 0.851 | 0.613 | 0.670 | 0.694 | 0.493 | 0.362 | 0.364 | 0.259 |
| | IQT-NAR (Cheon et al., 2021) | 0.908 | 0.906 | 0.728 | 0.802 | 0.860 | 0.624 | 0.680 | 0.707 | 0.499 | 0.372 | 0.372 | 0.269 |
| | CVRKD (Yin et al., 2022) | 0.913 | 0.917 | 0.748 | 0.829 | 0.872 | 0.655 | 0.691 | 0.733 | 0.501 | 0.416 | 0.413 | 0.287 |
| | Our NAR-teacher | 0.903 | 0.888 | 0.717 | 0.799 | 0.821 | 0.609 | 0.674 | 0.691 | 0.490 | 0.470 | 0.472 | 0.322 |
| NR-IQA | RKIQT (Ours) | 0.931 | 0.914 | 0.764 | 0.809 | 0.841 | 0.620 | 0.730 | 0.738 | 0.537 | 0.566 | 0.581 | 0.407 |

## 4.3 COMPARISON WITH SOTA IQA METHODS

Table 1 presents the comparative performance of the proposed RKIQT and other state-of-the-art NR-IQA methods, including convolution-based methods such as HyperNet (Su et al., 2020), as well as vision transformer-based methods such as DEIQT (Qin et al., 2023) and LoDa[2] (Xu et al., 2024). The evaluation results obtained from 8 diverse datasets demonstrate that RKIQT outperforms all other NR-IQA methods across each dataset. Notably, as shown in Table 9, RKIQT continues to benefit from larger backbone sizes, further demonstrating the effectiveness of our approach. Furthermore, Table 2 shows that our method outperforms various NAR-IQA approaches, including the teacher model, demonstrating the effectiveness of the distillation strategy in learning reference knowledge with fewer inputs. On the synthetic LIVE and TID2013 datasets, RKIQT achieves performance comparable to or better than FR-IQA methods like LPIPS. Although the results are not entirely superior, it is worth noting that the proposed method does not require reference images during inference, making it more suitable for real-world IQA tasks where reference images are unavailable.

## 4.4 GENERALIZATION CAPABILITY VALIDATION

We evaluate the generalization ability of RKIQT by employing a cross-dataset validation approach. In this approach, we train the NR-IQA model on one dataset and test it on others without fine-tuning or parameter adaptation. Table 3 shows the experimental results of SRCC averages on the five datasets. As observed, RKIQT achieves the best performance on five of the six cross-datasets. It clearly outperforms the other methods on the LIVEC dataset and shows competitive performance on the KonIQ dataset which strongly demonstrates the generalization ability.

Table 3: SRCC on the cross datasets validation. The best results are highlighted in bold, second-best is underlined.

| Training | LIVEFB | | LIVEC | KonIQ | LIVE | CSIQ |
|---|---|---|---|---|---|---|
| Testing | KonIQ | LIVEC | KonIQ | LIVEC | CSIQ | LIVE |
| DBCNN | 0.716 | 0.724 | 0.754 | 0.755 | 0.758 | 0.877 |
| P2P-BM | 0.755 | 0.738 | 0.740 | 0.770 | 0.712 | - |
| TReS | 0.713 | 0.740 | 0.733 | 0.786 | 0.761 | - |
| DEIQT | 0.733 | 0.781 | 0.744 | 0.794 | 0.781 | 0.932 |
| LoDa | 0.763 | 0.805 | 0.745 | 0.811 | - | - |
| RKIQT | 0.759 | 0.797 | 0.760 | 0.818 | 0.793 | 0.935 |

---

[2]For a fair comparison, we report the experimental results of LoDa with a ViT-B backbone (pre-trained on ImageNet-1k) on the KADID and KonIQ datasets.

Table 4: Ablation experiments on KADID, LIVEC, and KonIQ datasets (left) and MCD ablation experiments on LIVE, LIVEC, and KonIQ datasets (right). Bold entries indicate the best performance.

| Method | KADID | | LIVEC | | KonIQ | |
|---|---|---|---|---|---|---|
| | PLCC | SRCC | PLCC | SRCC | PLCC | SRCC |
| CNN-teacher | 0.865 | 0.866 | 0.892 | 0.866 | 0.921 | 0.903 |
| INN-teacher | 0.789 | 0.798 | 0.815 | 0.811 | 0.910 | 0.900 |
| NAR-teacher | 0.909 | 0.902 | - | - | - | - |
| baseline | 0.878 | 0.884 | 0.887 | 0.865 | 0.930 | 0.918 |
| w/o Regular. | 0.903 | 0.905 | 0.903 | 0.879 | 0.938 | 0.927 |
| w/o MCD | 0.902 | 0.902 | 0.907 | 0.881 | 0.939 | 0.926 |
| RKIQT | **0.911** | **0.911** | **0.917** | **0.897** | **0.943** | **0.929** |

| Method | LIVEC | | KonIQ | |
|---|---|---|---|---|
| | PLCC | SRCC | PLCC | SRCC |
| baseline | 0.887 | 0.865 | 0.930 | 0.918 |
| std | ±0.02 | ±0.017 | ±0.003 | ±0.004 |
| w/ DRD | 0.908 | 0.889 | 0.940 | 0.925 |
| std | ±0.008 | ±0.014 | ±0.002 | ±0.003 |
| w/ MCD | **0.917** | **0.897** | **0.943** | **0.929** |
| std | **±0.008** | **±0.009** | **±0.002** | **±0.002** |

## 4.5 Ablation Study

**Ablation on overall Distillation framework.** RKIQT consists of two main components: Masked Quality-Contrastive Distillation (MCD) and Inductive Bias Regularization. We conducted ablation studies to assess their individual contributions, as shown in Table 4. "w/o Regular." indicates MCD without Inductive Bias Regularization, and "w/o MCD" refers to regularization without MCD. The results demonstrate that both MCD and Inductive Bias Regularization significantly improve image quality representation, leading to the superior performance of RKIQT. Notably, our model outperforms the NAR-teacher model, which uses reference prior information. Specifically, the inductive bias regularization approach significantly improves the model's accuracy and stability, while the MCD technique has a more pronounced impact on the KADID dataset. This outcome is expected since inductive bias regularization involves a more expensive pre-training process, where each dataset is pre-trained with the corresponding teacher, introducing significantly more prior information than MCD. However, MCD still enables our model to achieve better performance and generalization than existing SOTA NR-IQA methods. In conclusion, the ablation studies confirm that both MCD and Inductive Bias Regularization are essential for improving model accuracy and stability.

Table 5: Ablation experiments for Generation Module in MCD on LIVEC dataset.

| Layers | Kernel | PLCC | SRCC |
|---|---|---|---|
| 1 | 3×3 | 0.907 | 0.885 |
| 2 | 3×3 | **0.917** | **0.897** |
| 3 | 3×3 | 0.909 | 0.886 |
| 2 | 5×5 | 0.916 | 0.894 |
| 3 | 5×5 | 0.909 | 0.888 |

Table 6: Performance of using different inductive bias teachers on LIVEC dataset.

| CNN | INN | PLCC | SRCC |
|---|---|---|---|
| | | 0.903 | 0.879 |
| ✓ | | 0.909 | 0.886 |
| | ✓ | 0.910 | 0.89 |
| ✓ | ✓ | 0.912 | 0.892 |
| ✓ | ✓ | **0.917** | **0.897** |

Table 7: Ablation experiments on Reverse Distillation (RD) on the LIVEC and KonIQ datasets.

| Method | LIVEC | | KonIQ | |
|---|---|---|---|---|
| | PLCC | SRCC | PLCC | SRCC |
| baseline | 0.894 | 0.875 | 0.935 | 0.922 |
| std | ±0.02 | ±0.017 | ±0.003 | ±0.004 |
| w/o RD | 0.911 | 0.886 | 0.941 | 0.928 |
| std | ±0.009 | ±0.014 | ±0.004 | ±0.003 |
| w/ RD | **0.917** | **0.897** | **0.943** | **0.929** |
| std | **±0.008** | **±0.009** | **±0.002** | **±0.002** |

**Ablation on Masked Quality-Contrastive Distillation.** To further investigate the effectiveness of MCD, we conducted ablation experiments where the feature distillation method was replaced with MCD and direct feature distillation (DRD), respectively. The results of these experiments, as shown in Table 4, indicate that the model trained using the MCD approach demonstrated significantly higher accuracy and stability across both synthetic and real-world datasets, particularly on the real-world dataset LIVEC. These findings clearly demonstrate that the MCD distillation method enhances the model's robustness in perceiving image distortions in natural environments. For a more detailed analysis of MCD, please refer to Sec. A.5 in the Appendix.

**Ablation on Inductive Bias Regularization and Reverse Distillation.** We compare test loss during training with the baseline (Fig. 3) to assess overfitting prevention. Regularization consistently leads to lower test errors, while the baseline shows higher errors and oscillations at 70/50 steps, indicating overfitting. In contrast, regularization maintains a steady reduction in errors. In the early phase of LIVEC training (before 35 steps), regularization has limited impact, but its benefits become evident after 70 steps through reverse distillation with inter-layer modules, gradually closing the gap between teacher and student models (Table 4). Over time, logit regularization effectively mitigates overfitting (Fig. 9). Additional experiments (as shown in Table 7) further confirm the role of reverse distillation

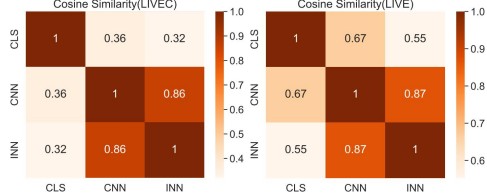

Figure 3: (a) and (b) are sensitivity experiment of hyper-parameters $\lambda_1$ and $\lambda_2$, Fig. 3(c) and (d) compare testing loss plots with regularization and baseline on LIVEC and KonIQ dataset which demonstrate the effectiveness of preventing overfitting

Table 8: Performance results using HQ reference images with different contents for knowledge distillation.

|  | TID2013 | | CSIQ | |
| --- | --- | --- | --- | --- |
| Reference Image | PLCC | SRCC | PLCC | SRCC |
| Content-align | **0.941** | **0.928** | **0.973** | **0.963** |
| Content-similar | 0.926 | 0.909 | 0.965 | 0.96 |
| Content-dissimilar | 0.917 | 0.897 | 0.97 | 0.958 |

Table 9: Performance results for various ViT sizes (pre-trained on ImageNet-1K).

| Encoder | PLCC | SRCC |
| --- | --- | --- |
| RKIQT(ViT-S) | 0.911 | 0.911 |
| RKIQT(ViT-M) | 0.917 | 0.914 |
| LoDa (ViT-B) | 0.920 | 0.912 |
| RKIQT(ViT-B) | **0.922** | **0.919** |

in improving performance and stability. The inter-layer modules help the student model learn more effectively from teachers with different inductive biases, such as in texture extraction and detecting subtle features (e.g., low contrast in Fig. 7). This strategy leverages prior knowledge and significantly enhances training efficiency. For more details on accelerated convergence, refer to Fig. 9.

**Ablation on Generation Block in MCD Module.** We tested various generation block configurations (Table 5). A single convolution layer showed minimal improvement, while three layers lowered performance compared to two. Additionally, 5x5 kernels increased computational cost without benefits over 3x3. Thus, we selected two convolution layers with one activation layer.

## 4.6 IN-DEPTH ANALYSIS

**The necessity of different inductive biases in inductive bias regularization.** We conducted ablation experiments on the use of INN and CNN networks, with results shown in Table 6. Performance declines when only one teacher is used, highlighting the necessity of both. This is in keeping with our previous observations that (1) INNs provide distinct inductive biases and output distributions compared to transformers, excelling on datasets like LIVE and CSIQ, while transformers, such as Musiq, perform better on others, as shown in Table 13 in the appendix. This diversity enriches the data perspectives for transformers. (2) Additionally, previous work relying solely on CNN teachers, like DeiT, suffered from increased bias-related errors. Introducing INNs helps balance these biases, reducing overfitting and improving model robustness. Please refer to Sec. A.5 for more details.

**Inductive Bias Token Enhances Perspective Diversity.**

To demonstrate that these tokens with different inductive biases indeed model unique features, we compute the cosine similarity between the CLS, CNN, and INN tokens of the distillation model (results are averaged over the LIVEC and LIVE datasets, respectively). As shown in Fig. 5, the result is between 0.32 and 0.7. This is significantly lower than the similarity between class and distillation labels in previous work (Touvron et al., 2021); 0.96 and 0.94 in DeiT-T and Deit-S, respectively. This confirms our hypothesis that modeling local and global features with multiple perspectives separately with separate tokens in Vits leads to a more comprehensive quality feature representation.

Figure 5: Cosine similarity between perceptual features of CLS token, CNN token, and INN token. The low similarity between them suggests that each token judges the image quality from a unique perspective.

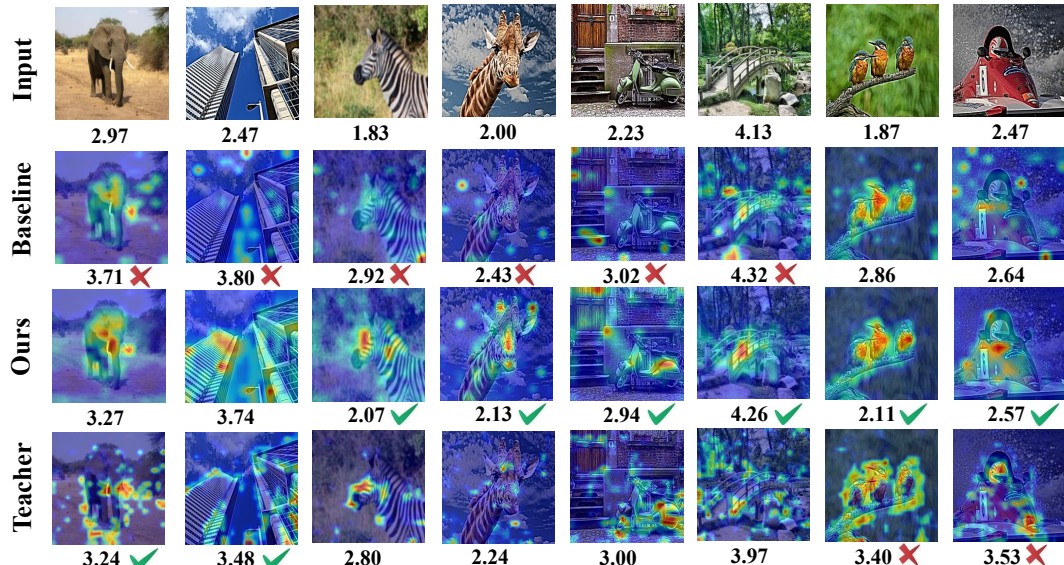

Figure 4: Activation maps of baseline, RKIQT, and NAR-Teacher using the Grad-CAM (Selvaraju et al., 2017). Mean Opinion Scores are displayed in the figures. Our RKIQT model is designed to focus more on image distortion and consequently improves image quality prediction performance. Red crosses indicate the worst predictions, while green checkmarks indicate the best predictions.

**Effect of Reference Image with Different Content.** Given that HQ images are randomly sampled, they may have no direct content relation to the LQ images. To assess the impact of this variability, we conducted additional experiments using content-aligned and content-similar HQ images. Content similarity was achieved through affine transformations (scaling: 0.95–1.05, rotation: $-5°$ to $5°$) (Liang et al., 2016). As shown in Table 8, the results show that using content-similar HQ images further improves model performance. However, even with content-dissimilar HQ images, our RKIQT still achieves superior results, demonstrating its robustness in learning from diverse reference knowledge.

**Visualization of quality attention map.** We use GradCAM (Selvaraju et al., 2017) to visualize feature attention maps (Fig. 4). The teacher model focuses on global edges rather than semantic information, emphasizing the importance of edges in image quality. In contrast, the baseline model often focuses on semantic content but is easily distracted, frequently attending to undistorted regions. RKIQT, benefiting from NR-IQA's semantic awareness and contrastive features learned from the teacher model, accurately identifies distorted areas. The prediction results indicate that RKIQT outperforms the baseline and teacher models across distortion levels, though distinguishing severe edge distortions remains challenging (first two columns) due to missing reference images. Nonetheless, RKIQT more accurately identifies distorted regions than the baseline.

**Analysis on Sensitivity of hyper-parameters.** In this paper, we use $\lambda_1$ and $\lambda_2$ in Eq. 6 to balance the MCD and regularization, respectively. In this subsection, we do the sensitivity study of the hyperparameters and conduct experiments on different Loss weights $\lambda$ to explore their effect on RKIQT. As shown in Fig. 3, the MCD and Inductive Bias Regularization are not very sensitive to the hyper-parameter $\lambda$, which is just used for balancing the loss. This indicates that the choice of hyper-parameter in our approach is relatively arbitrary, highlighting the robustness of our model.

## 5 CONCLUSION

The primary challenge for NR-IQA is the absence of effective reference information. To mitigate this issue, we introduce the reference knowledge into the NR-IQA and propose an RKIQT method. We make the first attempt to introduce human comparative thinking into the IQA model, thus ensuring a high consistency with the human subjective evaluation. In particular, we design a Masked Quality-Contrastive Distillation module that distills teachers' comparison knowledge given non-aligned high-quality images. Furthermore, an inductive bias regularization is proposed based on the CNN and INN networks. It allows the students with fewer inductive biases to learn from teachers with various inductive biases, and subsequently achieve a fast convergence and generalization capability. Experiments on 8 IQA datasets verify the superiority of the RKIQT.

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

# A APPENDIX / SUPPLEMENTAL MATERIAL

## A.1 APPENDIX OVERVIEW

The supplementary material is organized as follows: **Explanations for Concepts:** provide explanations for some of the concepts in the manuscript. **Training and Evaluation Details:** shows more training and evaluation details. **More ablation:** provides more ablation experiments, including MCD, random mask, Inductive Bias token, and Inter-layer. **Limitation:** analyzes the limitations of our RKIQT as well as directions for future work

## A.2 EXPLANATIONS FOR CONCEPTS

**ClS Token:** In classification tasks, the input image is divided into multiple patches. The Vision Transformer (ViT) model learns to extract global aggregate information by aggregating relationships between different patches through learnable CLS tokens.

**Token Inductive Bias:** This bias assigns different biases to tokens. The purpose is to align the inductive bias of tokens with that of teachers, enabling tokens to learn more effectively from their corresponding teachers.

**Expansion of INN:** Involution (INN) is a type of kernel that is shared across channels but distinct in spatial extent. INN exhibits precisely inverse inherent characteristics compared to convolution, enabling it to capture global spatial relationships in an image.

**Pixel-Aligned Reference:** This term refers to a clear reference image that corresponds to a distorted image, having exactly the same content information as the distorted image.

**Offline Distillation:** During training, knowledge from a pre-trained teacher model is transferred to a student network. Only the student network is trained, while the parameters of the teacher are frozen.

**Non-aligned Reference:** In this paper, "aligned" refers to situations where a blurred image has a corresponding clear image of the same version. For instance, if we have a blurry photo due to camera shake, two images are considered aligned when the camera captures the distorted image's corresponding clear image under the same scene, view angle, and lighting conditions. However, obtaining this aligned clear image is often challenging in practical settings. Typically, the reference image used is non-aligned. Therefore, the term "non-aligned model" means that the pixel of low-quality image and the high-quality image don't have a one-to-one correspondence. In other words, the high-quality image only needs to be clear, while the image content can vary.

## A.3 TRAINING AND EVALUATION DETAILS

Our teacher models are both pre-trained and freeze parameters during student training.

**Implementation Details:** To train the student network PyTorch (Paszke et al., 2019), we follow the typical strategy of randomly cropping the input image into 10 image patches with a resolution of $224 \times 224$. Each image patch is then reshaped as a sequence of patches with a patch size of p = 16 and a dimension of input tokens as in $D = 384$. We create the Transformer encoder based on the ViT-S proposed in DeiT III (Touvron et al., 2022), with the encoder depth set to 12 and the number of heads h = 6. The depth of the decoder is set to 1. The model is trained for 9 epochs with a learning rate of $2 \times 10^{-4}$ and a decay factor of 10 every 3 epochs. The batch size varies depending on the size of the dataset, with a batch size of 16 for LIVEC and 128 for KonIQ. For each dataset, 80% of the images are used for training, and the remaining 20% are used for testing. We repeat this process 10 times to mitigate performance bias and report the average of SRCC and PLCC. For our pre-trained CNN, INN teacher, and NAR-teacher, the pre-training follows a similar method to student training, with hyperparameters from previous work (Qin et al., 2023).

**Training Stage:** As depicted in Fig. 2 and algorithm 1, begins with an input image. The student model, along with three different inductive bias tokens, and the NAR-teacher model, acquire both LQ features and the difference in distribution between HQ and LQ features. To improve the student's feature representation, we employ Mask Quality Contrast distillation. This involves masking the student's feature map and generating a new feature using a simple generation module. The generation process is supervised by the NAR-teacher's differential features. Subsequently, the student's three

different inductive bias tokens enter the decoder to predict three quality scores. Each quality score is supervised by a specific inductively biased teacher. However, instead of directly using the teacher logits with different inductive biases to supervise the students, we introduce a learnable intermediate layer. This is done to mitigate the potential large quality perception gap between teachers and students. Additionally, it is worth noting that the learnable intermediate layer is supervised by both the students and the CNN and INN teachers.

**Inference Stage:** All teacher models, feature distillation, and regularization techniques are no longer utilized. In other words, as depicted in algorithm 2, the student model is directly applied for inference without reference images or high-quality images.

---

**Algorithm 1** Training Process of RKIQT

---

**Require:**
1: Low-quality (LQ) images: $X_{LQ}$
2: LQ images' ground truth: $Y_{gt}$
3: High-quality (HQ) images: $X_{HQ}$
4: Inductive Bias Student Network: $S$
5: CNN teacher's learnable intermediate layer: $T_{cnn}^l$
6: INN teacher's learnable intermediate layer: $T_{inn}^l$
7: Encoder layer number $i$, $1 \leq i \leq L$
8: Non-aligned reference teacher (NAR-teacher): $T_{nar}$
9: Pre-trained CNN teacher: $T_{cnn}$, INN teacher: $T_{inn}$
10: Loss hyper-parameters: $\lambda_1$, $\lambda_2$
11: **Masked Quality-Contrastive Distillation:**
12: **for** each encoder layer $i = 1, 2, ..., L$ **do**
13:     Obtain $F_{LQ}$ of input $X_{LQ}$ using the $S$ encoder.
14:     Obtain LQ-HQ difference-aware features $F_{HQ-LQ}$ using the $T_{nar}$ encoder.
15:     Randomly mask $F_{LQ}$ to obtain $F_{mask}$.
16:     Generation module to restore $F_{mask}$ to the $F_{new}$.
17:     MSE loss between $F_{new}$ and $F_{HQ-LQ}$: $\mathcal{L}_{fea}^i$.
18: **end for**
19: Sum up $\mathcal{L}_{fea}^i$ of all layers.
20: **Inductive Bias Regularization:**
21: Get the output $Y_{cls}, Y_{s_{cnn}}, Y_{s_{inn}}$ using the $S$.
22: Obtain $Y_{T'_{cnn}}, Y_{T'_{inn}}$ of input $X_{LQ}$ using $T_{cnn}$ and $T_{inn}$.
23: Obtain pseudo-label $Y_{T_{cnn}}, Y_{T_{inn}}$ of input $X_{LQ}$ using $T_{cnn}^l$ and $T_{inn}^l$, respectively.
24: Calculate loss $\mathcal{L}_{logits}$ in Equ. 3,5 of our manuscript.
25: Calculate loss $\mathcal{L}_{all}$ in Equ. 6 of our manuscript.
26: Use $\mathcal{L}_{all}$ to update $S$.
    **Output:** $S$

---

**Algorithm 2** Inference Process of RKIQT

---

**Require:**
1: Low-quality (LQ) images: $X_{LQ}$
2: Inductive Bias Student: $S$
3: **Testing Process:**
4: Using ClS token, Conv token, and Inv token in the $S$ to get the quality score $Y_{cls}, Y_{s_{cnn}}, Y_{s_{inn}}$.
5: Select $Y_{cls}$ as the final output.
    **Output:** $Y_{cls}$

---

A.4   REFERENCE-GUIDED TRANSFORMER STUDENT DECODER

As mentioned before, we propose cross-inductive bias teachers that can focus on various inductive biases (Sec. 3.3) to achieve fast convergence and prevent overfitting. To align additional learnable tokens with different inductive bias teachers, we introduce token inductive bias alignment. We use

Table 10: Comparison of model complexity during training and inference phases.

| Phase | Param (M) | GFLOPs | Memory (GB) | Throughput (pairs/s) |
|---|---|---|---|---|
| Training | 32M | 32.71 | 15.6 | 126.8 |
| Inference | 28M | 7.78 | 1.76 | 388.7 |

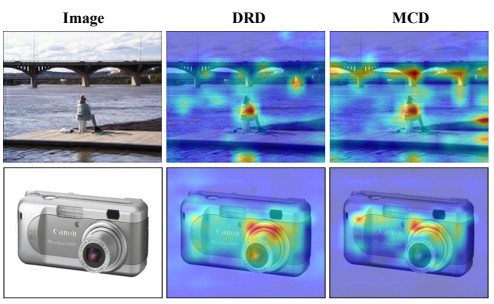

Figure 6: Images from left to right are the input images and their attention maps with DRD and MCD. As observed, MCD pays more attention to the background distortion region, and to the quality distortion region of the subject.

Table 11: MCD ablation experiments on LIVE, LIVEC, and KonIQ datasets. Bold entries indicate the best performance.

| | LIVE | | LIVEC | | KonIQ | |
|---|---|---|---|---|---|---|
| Method | PLCC | SRCC | PLCC | SRCC | PLCC | SRCC |
| baseline | 0.978 | 0.977 | 0.887 | 0.865 | 0.930 | 0.918 |
| std | ±0.004 | ±0.005 | ±0.02 | ±0.017 | ±0.003 | ±0.004 |
| w/ DRD | 0.983 | 0.981 | 0.908 | 0.889 | 0.940 | 0.925 |
| std | ±0.003 | ±0.003 | ±0.008 | ±0.014 | ±0.002 | ±0.003 |
| w/ MCD | **0.986** | **0.984** | **0.917** | **0.897** | **0.943** | **0.929** |
| std | **±0.002** | **±0.003** | **±0.008** | **±0.009** | **±0.002** | **±0.002** |

three tokens: Class token, Conv token, and Inv token. We apply truncated Gaussian initialization to the Class token to eliminate its inductive bias and align it with the ground truth (Touvron et al., 2021). On the other hand, we introduce the corresponding inductive bias into the remaining two tokens. The Conv token and Inv token use the average pooling outputs of convolution stem and involution stem, respectively, with added position embeddings. The output of the encoder includes three inductive bias tokens denoted by $\hat{F}_o \in \mathbb{R}^{3 \times D}$. Then, we follow previous work (Qin et al., 2023) by introducing a transformer decoder to further decode inductive biases CLS, Conv, and Inv tokens through multi-head self-attention (MHSA), thus making the extracted features more significant and comprehensive to the image quality. The queries $Q_d$ of the decoder are written by:

$$Q_d = \text{MHSA}(\text{Norm}(\hat{F}_o + J)) + (\hat{F}_o + J), \tag{7}$$

where $J \in \mathbb{R}^{3 \times D}$ is initialized with random numbers, which evaluate the image quality from different perspectives (Qin et al., 2023).

$$\hat{Y} = \text{MLP}(\text{MHCA}(\text{Norm}(Q_d), K_d, V_d) + Q_d) \tag{8}$$

During Multi-Head Cross-Attention (MHCA), we utilize $Q_d$ to interact with the features of the image patches preserved in the encoder outputs. The results are then fed to an MLP to derive the final quality score $\hat{Y}$. The transformer decoder can significantly improve the learning ability of the ViT-based NR-IQA model, thus improving the performance of the model and generalization ability. We further present a comparison of the training and inference complexity of the RKIQT, as shown in Table 10.

## A.5 MORE ABLATION

**Ablation on Masked Quality-Contrastive Distillation.**

To further investigate the effectiveness of the proposed MCD, we conduct ablation experiments to train the model by changing the way of feature distillation to MCD and direct feature distillation (DRD), respectively. We repeat the experiment 10 times for each set of training data and report the average of PLCC, and SRCC. The experimental results are detailed in Table 4. Training model via MCD achieves the best accuracy compared to DRD on both synthetic and authentic datasets, especially on the authentic dataset LIVEC These observations vividly show that the distillation way of MCD enhances the robustness of the model to image distortion perception in natural environments. In other words, RKIQT effectively utilizes the information of the asymmetric reference graph and achieves the best performance on both synthetic and real datasets.

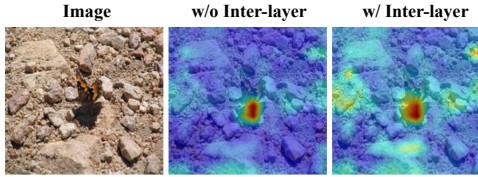

Figure 7: The first picture is the distorted picture. The remaining images are the attention map without and with learnable Inter-layer, respectively. Incorporating the Inter-layer, our model pays more attention to the quality-aware features.

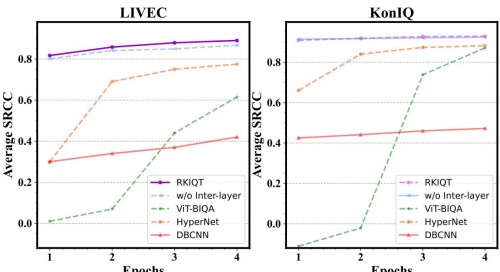

Figure 9: Average SRCC versus Epochs on different datasets ablation on Inductive Bias Regularization.

Figure 8: Inter-layer ablation experiments on LIVEC and KonIQ datasets. Bold entries indicate the best performance.

| Method | LIVEC | | KonIQ | |
|---|---|---|---|---|
| | PLCC | SRCC | PLCC | SRCC |
| baseline | 0.894 | 0.875 | 0.935 | 0.922 |
| std | ±0.02 | ±0.017 | ±0.003 | ±0.004 |
| w/o Inter-layer | 0.911 | 0.886 | 0.941 | 0.928 |
| std | ±0.009 | ±0.014 | ±0.004 | ±0.003 |
| w/ Inter-layer | **0.917** | **0.897** | **0.943** | **0.929** |
| std | **±0.008** | **±0.009** | **±0.002** | **±0.002** |

Table 12: Mask function ablation experiments on LIVEC datasets. Bold entries indicate the best performance.

| Method | PLCC | SRCC |
|---|---|---|
| RKIQT w/ random mask | 0.917 | 0.897 |
| RKIQT w/ Gaussian (center) | 0.916 | 0.897 |
| RKIQT w/ all mask (center) | 0.916 | 0.896 |
| RKIQT w/ Gaussian (edge) | **0.919** | 0.896 |
| RKIQT w/ all mask (edge) | 0.918 | **0.900** |

We provide a detailed analysis and consider that (i) MCD aids the model in acquiring HQ-LQ distribution difference knowledge (i.e., contrastive ideas) and (ii) MCD preserves both local distortion and global semantic features in the masked pixels, in conjunction with (i) to generate more comprehensive quality-aware features. It is important to note that HQ-LQ distribution difference knowledge is mainly represented by the edge of foreground and background in visualization, as illustrated in row 4 of Fig. 4 of the manuscript. This is further demonstrated in Fig.6, which presents images containing complex content (top) and simple content (bottom), accompanied by the corresponding student encoder visualization outcomes. When the image is relatively simple, MCD's response to background quality perception is significantly reduced, with greater focus placed on the distortion of the foreground content, thus confirming the second point (ii). However, as the complexity of the image scene increases, MCD also starts to respond more to the quality perception of the edge background, thus supporting the first point (i).

**The effectiveness of accelerating convergence.** To demonstrate the effect of the regularization on convergence, we evaluate the training efficiency and performance of RKIQT distillation, as shown in Fig. 9, which depicts the SRCC with an increasing number of epochs on LIVEC and KonIQ test sets. The results show that RKIQT converges significantly faster than the other methods, achieving the fastest convergence after only one epoch of training, which outperforms the second-best NR-IQA method in Table 1 of the manuscript. Furthermore, on LIVEC, the use of the Inter-layer module greatly reduces the negative impact of the teacher network's less ideal performance, indicating that the Inter-layer module preserves the diversity of knowledge and accelerates convergence. These observations demonstrate that RKIQT and the teacher can "learn from each other", with the teacher adapting its teaching to the student's abilities, resulting in more comprehensive knowledge and significantly improved model stability.

**Ablation on Random Mask.** Given that local distortions are often concentrated in the foreground or center regions of an image, we conducted four sets of experiments to investigate the effects of local distortion erasure, as shown in Table. 12. These experiments focused on the center and edge regions of the image. 1)RKIQT W/ Gaussian(center): The random mask function was replaced with a Gaussian distribution probability mask function, and the central region of the feature map was masked with a higher probability. 2)RKIQT W/ Gaussian(edge): The random mask function was replaced with a Gaussian distribution probability mask function, and the edge region of the feature

| | LIVE | | CSIQ | | TID2013 | | KADID | |
|---|---|---|---|---|---|---|---|---|
| | PLCC | SRCC | PLCC | SRCC | PLCC | SRCC | PLCC | SRCC |
| CNN | 0.957 | 0.958 | 0.937 | 0.931 | 0.883 | 0.866 | 0.865 | 0.866 |
| INN | 0.965 | 0.963 | 0.948 | 0.939 | 0.901 | 0.901 | 0.789 | 0.798 |
| MUSIQ | 0.911 | 0.940 | 0.893 | 0.871 | 0.815 | 0.773 | 0.872 | 0.875 |
| | LIVEC | | KONIQ | | SPAQ | | LIVEFB | |
| | PLCC | SRCC | PLCC | SRCC | PLCC | SRCC | PLCC | SRCC |
| CNN | 0.892 | 0.866 | 0.921 | 0.903 | 0.864 | 0.860 | 0.653 | 0.557 |
| INN | 0.815 | 0.811 | 0.910 | 0.900 | 0.911 | 0.914 | 0.572 | 0.521 |
| MUSIQ | 0.746 | 0.702 | 0.928 | 0.916 | 0.928 | 0.918 | 0.661 | 0.566 |

Table 13: Performance comparison of NR-IQA methods with different inductive biases.

map was masked with a higher probability. 3)RKIQT W/ all mask(center): In this experiment, all blocks in the central region were masked, while the edge region was masked with a lower probability. 4)RKIQT W/ all mask(edge): In this experiment, all blocks in the edge region were masked, while the central region was masked with a lower probability.

From the experimental results shown in Table 12, we conducted two sets of experiments to mask the central region. Interestingly, the experimental results indicate that masking the central region had almost no impact on the performance of our model. On the contrary, when we considered applying a larger probability of masking to the edge region or even masking the entire image except for the central region, we observed some improvement in the model's performance. These findings suggest that the erasure of local distortions has little effect on the model's performance, and in some cases, an appropriate masking mechanism can even enhance the model's performance. This provides a potential direction for our future work.

**The necessity of CNN teacher and INN teacher in inductive bias regularization.** Intuitively, including the INN teacher is necessary for three key reasons: 1) Highlighting Different Data Patterns: Previous studies (Li et al., 2021) have shown that INN and CNN focus on different data patterns due to their opposing inductive biases. As shown in Table 13, INN performs better on datasets like LIVE, CSIQ, TID, and SPAQ, while CNN excels on others. Teachers with different inductive biases provide complementary data perspectives, leading to more accurate and comprehensive representations. Using both helps the transformer learn a more complete data representation (Zhang et al., 2023a; Pan et al., 2022). 2) Mitigating CNN Biases: Previous studies (Zhao et al., 2023a) have shown that DeiT relying solely on CNN teachers results in a strong influence from CNN inductive biases, which can increase classification errors. Introducing an INN teacher with opposing inductive biases can balance this effect, reducing the impact of specific CNN biases and alleviating related negative regularization. 3) From empirical evidence, we conducted ablation experiments on the use of INN and CNN networks. The results, as shown in the Table 6, indicate that performance declines when only one of the teachers is used. This demonstrates that both CNN and INN teachers are indispensable for optimal performance.

## A.6    LIMITATION

Although we have demonstrated the superiority of RKIQT and found that incorporating random and non-aligned reference information into traditional no-reference image quality assessment is highly beneficial, there remains an important issue that cannot be ignored. Specifically, there may be limitations for certain tasks such as underwater images and medical images, because the quality contrast knowledge (e.g., shift and artifacts) is quite different from those in traditional NR-IQA metrics (e.g., noise, and compression). Therefore, exploring how to adapt this framework to similar directions in the future is an interesting area for further investigation.

## A.7    BROADER IMPACTS

This paper aims to improve Image Quality Assessment (IQA) and considers its potential societal impacts. Enhanced IQA models can significantly improve user experiences on digital platforms by

ensuring that high-quality images are displayed. This is especially beneficial in online retail, social media, and digital advertising, where the quality of visual content greatly influences user engagement and satisfaction. However, IQA models may be vulnerable to adversarial attacks. Malicious actors might manipulate image quality ratings to deceive users or automated systems. For instance, low-quality advertisement images could be falsely rated as high quality, misleading consumers and reducing the effectiveness of advertising campaigns. To mitigate these risks, a possible strategy is to implement monitoring systems to detect and respond to anomalies will help maintain the reliability and integrity of IQA models.

