# OpenReview forum: "Less is More: Learning Reference Knowledge Using No-Reference Image Quality Assessment"
_ICLR.cc/2025/Conference — Submitted to ICLR 2025_

### Official Review · Reviewer_Qsu4 · 2024-10-22

**Soundness:** 3
**Presentation:** 2
**Contribution:** 2
**Rating:** 5
**Confidence:** 5

**Summary:**

This work aims to enhance NR-IQA by leveraging comparison knowledge used in FR-IQA but without relying on reference images during inference time. Specifically, a Masked Quality-Contrastive Distillation method is designed to distill comparison knowledge from a non-aligned reference (NAR) teacher model to an NR student model. In addition, an inductive bias regularization method is proposed to learn knowledge from both a CNN teacher and an Involution Neural Network (INN) teacher, aiming to integrate complementary inductive biases into the backbone ViT.

**Strengths:**

+ This work is well-motivated, leveraging comparison knowledge to NR-IQA models without accessing the reference images during inference is a promising direction.
+ Distilling inductive bias to a ViT backbone is conceptually reasonable.
+ The resulting model attains high performance on multiple IQA benchmarks.

**Weaknesses:**

- To the Reviewer, this work is more a good-engineered solution than an academic finding. Despite the superior final performance, no new technique is proposed.
- The results of the compared methods seem to be directly copied from their corresponding papers. Different splits of train/val/test may significantly affect the fairness of performance comparison.
- Ablation studies are conducted on LIVEC only or LIVEC and KoNIQ, which may not be representative enough.

**Questions:**

There is no validation set. How to pick the epoch?

---

> ### Author Response · Authors · 2024-11-23
> **Author Responses**
>
> > **W1**: To the Reviewer, this work is more a good-engineered solution than an academic finding.  Despite the superior final performance, no new technique is proposed.
>
> **(1)** We would like to clarify that our technical innovations are reflected in two key aspects:
>
> - Unlike previous KD methods, our MCD approach does not directly align all pixels with the teacher's features. Instead, **MCD masks certain pixels, training different local pixels to perceive quality differences**. MCD not only reduces the negative effects (inconsistency between HQ-LQ features and LQ features) of traditional MSE loss but also significantly enhances the quality perception capabilities of local pixels.
>
> - Unlike traditional logits-based distillation, **we use a learnable intermediate layer for reverse distillation to bridge the gap** between teachers and students with different inductive biases, enabling better learning of both local distortions and global quality perception.
>
> **(2)** Additionally, it is crucial to emphasize our **major contribution: introducing comparative awareness into NR-IQA**. The introduction of comparative awareness aligns more closely with human perception and is expected to offers a general framework that is orthogonal to existing NR-IQA architectures.
>
> Thank you for your comments. We will emphasize these contributions in the final draft.
>
> > **W2**: The results of the compared methods seem to be directly copied from their corresponding papers.  Different splits of train/val/test may significantly affect the fairness of performance comparison.
>
> Our comparison is relatively fair. **Regarding the results of competing models, we ensure a fair comparison by selecting the higher performance between publicly available results and those reproduced under the same data splits.**
> Specifically, we compared our testing outcomes with publicly available implementations. For instance, HyperNet achieved a PLCC of 0.876 on the LIVEC dataset (compared to the publicly reported 0.882), DEIQT achieved a PLCC of 0.885 (compared to 0.894), and TreS achieved a PLCC of 0.866 (compared to 0.877). Ultimately, we report the higher performance from either our reproduction or the publicly available results to ensure fairness and transparency in the comparison.
>
> > **W3**: Ablation studies are conducted on LIVEC only or LIVEC and KoNIQ, which may not be representative enough.
>
> Thank you for your valuable feedback. Due to space constraints, we selected the most representative authentic datasets, LIVEC and KonIQ. Based on your suggestions, we have further supplemented the ablation experiments on synthetic and authentic datasets, as shown in the table below. The results demonstrate that MCD facilitates the learning of comparative information. However, due to the limited inductive biases in ViT, it may be prone to overfitting. Combining MCD with regularization achieves the best overall performance, maximizing the effectiveness of the approach.
>
> | **Method**       | **KADID (PLCC)** | **KADID (SRCC)** | **CSIQ (PLCC)** | **CSIQ (SRCC)** | **LIVE (PLCC)** | **LIVE (SRCC)** | **SPAQ (PLCC)** | **SPAQ (SRCC)** |
> |-------------------|------------------|------------------|------------------|------------------|------------------|------------------|------------------|------------------|
> | Baseline          | 0.878           | 0.884           | 0.952           | 0.935           | 0.978 | 0.977 |  0.920   |  0.914   |
> | w/o Regular.      | 0.903           | 0.905           | 0.959           | 0.947          | 0.980    |  0.977   | 0.923   | 0.919    |
> | w/o MCD           | 0.902           | 0.902           | 0.961          | 0.949           | 0.981    | 0.978    | 0.925   | 0.918   |
> | **RKIQT**         | **0.911**       | **0.911**       | **0.970**       | **0.958**       | **0.986**   | **0.984**   |  **0.928**    |  **0.923**   |
>
>
>
> > **W4**:  There is no validation set. How to pick the epoch?
>
> Thank you for your feedback. Following LoDa and DEIQT methods, we pick the last epoch.

---

> > ### Comment · Reviewer_Qsu4 · 2024-11-26
> >
> > Thanks for the response. The rebuttal does not convince me about the technical novelty and experimental results. All the claims from the authors about the advantage of the proposed MCD are not rigorously verified. Regarding the experimental results, the standard should be unified across different datasets and methods. Reporting the higher results has nothing to do with fairness.
> >
> > Based on the comments, I cannot change my rating.

---

> ### Author Response · Authors · 2024-11-26
> **Deadline reminder and looking forward to discussion**
>
> Dear Reviewer Qsu4,
>
> We deeply appreciate your efforts in reviewing our paper, and we are particularly grateful for the invaluable suggestions that have greatly enhanced its quality.
>
> In response, we have clarified the technical contributions of our method **(W1)**, explained the fairness of our method comparisons **(W2)**, and conducted additional ablation studies across multiple datasets **(W3)**. Furthermore, we have provided an explanation regarding the selection of epochs **(Q1)**.
>
> We kindly request that you review our rebuttal and let us know if further clarification is needed. Your insights and suggestions are crucial for improving our work. Thank you for your time and dedication!
>
> Best regards,
>
> Authors

---

### Official Review · Reviewer_zofg · 2024-10-30

**Soundness:** 3
**Presentation:** 2
**Contribution:** 2
**Rating:** 3
**Confidence:** 5

**Summary:**

The authors introduce a framework called RKIQT, which aims to learn reference knowledge for image quality assessment without needing aligned reference images. The proposed method includes a Masked Quality-Contrastive Distillation (MCD) approach and an inductive bias regularization strategy. The authors claim that their method outperforms existing NR-IQA methods across multiple datasets.

**Strengths:**

*  The introduction of MCD to distill knowledge from non-aligned references is a new idea that could potentially enhance NR-IQA.
*  The authors conduct experiments across eight standard IQA datasets, demonstrating a commitment to empirical validation.

**Weaknesses:**

* It is unclear to me how adjusting the inductive biases of the ViT ensures rapid convergence and prevents overfitting.

* The authors claim that "our method utilizes less input—eliminating the need for reference images during inference—while achieving better performance than some IQA methods that do require reference images." However, the results show that the proposed model underperforms when compared to FR-IQA models (DISTS and IQT). Could you clarify this discrepancy?

* Why is this model considered the first attempt to transfer more HQ-LQ difference priors and rich inductive biases to NR-IQA via knowledge distillation (KD) in comparison to other KD-based models, such as the one proposed by Yin et al. (2022)? The novelty of using the KD scheme for IQA is limited.

* For the non-aligned reference teacher, directly computing the feature differences between HQ and LQ images is physically meaningless. Additionally, there is no ablation study to verify the effectiveness of the reference image.

**Questions:**

* There is no validation set used for model training. How do the authors conduct model selection and hyperparameter tuning? Using the test set for these purposes may violate established machine learning practices.

* The model should assess generalization by training on authentic distortions and testing on synthetic distortions (or vice versa).

* Why is there a discrepancy in SRCC performance on TID2013 between Table 1 and Table 8?

* In Line 478, the reported result of 0.86 falls outside the range of 0.32 to 0.7. Please clarify.

* In Fig. 4, the activation maps fail to focus on image distortions. For example, in the last two columns, the background is full of artifacts, but only the birds are highlighted.

---

> ### Author Response · Authors · 2024-11-23
> **Author Responses (1/n)**
>
> > **W1**: It is unclear to me how adjusting the inductive biases of the ViT ensures rapid convergence and prevents overfitting.
>
> We sincerely apologize for any confusion caused. Adjusting inductive biases enhances the student model's ability to capture both global and local representations of the distorted image, ensuring effective prevention of overfitting and achieving faster convergence. Below, we provide a detailed explanation from both theoretical and empirical perspectives:
> - **Enhanced Representation via Complementary Inductive Biases**
>   - Models with different inductive biases—such as CNNs and INNs—tend to focus on distinct data patterns. For example, as shown in Table 13, INNs excel on datasets such as LIVE, CSIQ, TID, and SPAQ, while CNNs perform better on others. By using teachers with diverse inductive biases, we provide complementary perspectives to the student model, enabling it to comprehensively learn both global and local features of distorted image, which is critical for rapid convergence and reduced overfitting [1, 2].
>
> - **Empirical Validation of Regularization**
>   - The incorporation of Regularization significantly reduces test error oscillations, as shown in Fig. 3 of the paper, showcasing its effectiveness.
>
> - **Empirical Validation of Rapid Ccnvergence**
>   - As shown in Fig. 9, RKIQT converges significantly faster than other methods, reaching peak convergence within a single training epoch. This highlights its efficiency and effectiveness in improving training performance.
>
> ---
> References
>
> [1] ViTAEv2: Vision Transformer Advanced by Exploring Inductive Bias for Image Recognition and Beyond. IJCV2023
>
> [2] On the Integration of Self-Attention and Convolution for Improved Vision Models. CVPR2022
>
> > **W2**: (1) The claims do not match the experimental results. (2) Could you clarify this discrepancy?
>
> - (1) Thank you for your valuable feedback. We would like to clarify that while the proposed method may underperform certain FR-IQA methods, RKIQT demonstrates superior performance compared to most non-aligned reference-based IQA (NAR-IQA) methods, which supports our claim.
> - (2) We attribute the performance gap with FR-IQA methods to lack of the stable HQ-LQ difference information during training. As shown in Table 8 of the manuscript, when original reference images are included in the training process, RKIQT's performance on the TID2013 dataset improves significantly, becoming competitive with FR-IQA methods like DISTS and LPIPS. This demonstrates the critical role of stable HQ-LQ difference features in enhancing RKIQT's performance and highlights the potential for improvements through the use of more effective reference images in the future work.
>
> |Method|LIVE(PLCC)|LIVE (SRCC)|CSIQ (PLCC)| CSIQ (SRCC) | TID2013 (PLCC)  | TID2013 (SRCC) |
> |-|-|-|-|-|-|-|
> |LPIPS| 0.978  | 0.972  | 0.970  | 0.967  | 0.944  | 0.936  |
> |DISTS| 0.980  | 0.975  | 0.973  | 0.965  | 0.947  | 0.943 |
> |RKIQT w/ FR | 0.986   | 0.984 |  0.973 | 0.963 |  0.941 |0.928 |
>
> > **W3**: The novelty of using KD for IQA seems limited.
>
> Thank you for your feedback. Let us outline the differences between our proposed KD approach and CVRKD-IQA's:
>    - To clarify, **RKIQT operates in an NR-IQA setting**, meaning it does not require additional reference images during inference. In contrast, **CVRKD-IQA belongs to NAR-IQA** and still depends on high-quality reference images for its evaluation.
>    - **A key distinction lies in the KD strategies used.** In CVRKD-IQA, the student model is designed to extract the difference features between distorted and reference images, making direct alignment (MSE loss) with the teacher’s difference features feasible. However, for NR-IQA, the student model only has access to distorted images, which are inherently **inconsistent with the HQ-LQ distribution features captured by the teacher.** Directly mimicking the teacher’s output in this case can lead to negative regularization effects, reducing both performance and stability (see Table 4).
>    - To address this issue, our MCD approach **does not directly align all pixels** with the teacher’s features. Instead, MCD masks certain pixels and trains different local pixels to perceive quality differences. This strategy mitigates the negative effects of traditional MSE loss (caused by inconsistency between HQ-LQ features and LQ features) and significantly enhances the quality perception capabilities of local pixels.
>    - Additionally, we introduce a KD-based regularization method to reduce overfitting (e.g., mitigating the impact of changes in reference image content in MCD) during training by efficiently capturing both local distortions and global features of the distorted image. Specifically, the method integrates complementary inductive biases from CNNs and INNs into the ViT framework. A learnable intermediate layer further bridges the inductive bias gap between teacher and student models, improving the model's ability to process distorted images.

---

> ### Author Response · Authors · 2024-11-23
> **Author Responses (2/n)**
>
> > **W4 (1)**: Directly computing feature differences between HQ and LQ images for the non-aligned reference teacher lacks physical meaning.
>
> (1) Thank you for your insightful question. The Free Energy Principle[1,2] states that a human visual system (HVS) minimizes free energy (F) to reduce uncertainty and improve perceptual prediction accuracy. Based on this principle, our NAR-teacher computes the feature differences between HQ and LQ images, effectively reducing posterior distribution uncertainty and providing reliable reference knowledge. Detailed theoretical and experimental evidence is presented below:
>
> ---
>
> **1. Free Energy Principle Overview**
> - The Free Energy Principle states that HVS minimizes **free energy (F)** to reduce uncertainty and improve prediction accuracy. Free energy is defined as:
>
> \\[
> F = D_{KL}(q(s) \| p(s | o)) - \log p(o),
> \\]
>
> - where:
>   - \\(q(s)\\): The internal predicted distribution of the true state \\(s\\) (e.g., scene features).
>   - \\(p(s | o)\\): The true posterior distribution conditioned on the observed data \\(o\\) (e.g., image features).
>   - \\(-\log p(o)\\): A constant term representing the marginal likelihood of observations \\(o\\).
>
> - Free energy can be approximated as:
> \\[
> F \approx D_{KL}(q(s) \| p(s | o)),
> \\]
> - where reducing \\(D_{KL}\\) corresponds to minimizing uncertainty.
> ---
> **2. Impact of Adding a High-Quality Reference Image**
>
> - With only the LQ image \\(o_d\\), the free energy is:
> \\[
> F_{\text{no ref}} \approx D_{KL}(q(s) \| p(s | o_d)).
> \\]
>
> - Adding an HQ reference image \\(o_r\\) changes the observation to \\(o = \{o_d, o_r\}\\), and the free energy becomes:
> \\[
> F_{\text{with ref}} \approx D_{KL}(q(s) \| p(s | o_d, o_r)).
> \\]
>
> - We aim to show:
> \\[
> F_{\text{with ref}} < F_{\text{no ref}}.
> \\]
> ---
> **3. Why Does Adding a Reference Image Reduce Free Energy?**
>
> - 1. **Mathematical Justification**:
>    - Adding the HQ reference image \\(o_r\\) changes the posterior distribution to:
>   \\[
>   p(s | o_d, o_r) = p(s | o_r)p(o_r | o_d),
>   \\]
>   - where:
>     - \\(p(s | o_r)\\): Encodes state information directly from the HQ reference.
>     - \\(p(o_r | o_d)\\): Encodes the relationship between the HQ and LQ images.
>
>   - **In practice, \\(p(o_r | o_d)\\) is difficult to compute directly, so it is approximated using feature differences \\(f(o_r) - f(o_d)\\)**, where \\(f(\cdot)\\) is the feature extractor. These differences capture distortions between HQ and LQ images, serving as a proxy for their relationship. Substituting this approximation into the posterior distributions, we have:
>
>       \\[
>       p(s | o_d, o_r) \propto p(s)p(f(o_r) - f(o_d) | s),
>       \\]
>       \\[
>       p(s | o_d) \propto p(s)p(f(o_d) | s),
>       \\]
>       - where:
>         - \\(p(s)\\): The prior distribution of the true state \\(s\\).
>         - \\(p(f(o_r) - f(o_d) | s)\\): The likelihood of the feature differences given the state \\(s\\).
>         - \\(p(f(o_d) | s)\\): The likelihood of the LQ image features given the state \\(s\\).
>   - The term \\(p(f(o_r) - f(o_d) | s)\\) reduces the uncertainty of \\(p(s | o_d, o_r)\\) compared to \\(p(s | o_d)\\) through two key aspects:
>     - **Lower Entropy:** The HQ information in \\(o_r\\) ensures that \\(p(f(o_r) - f(o_d) | s)\\) has lower entropy compared to \\(p(f(o_d) | s)\\), concentrating the probability mass around the likely values of \\(s\\).
>     - **Additional Constraints:**  The feature differences \\(f(o_r) - f(o_d)\\) impose additional constraints on \\(s\\), refining the posterior \\(p(s | o_d, o_r)\\) and making it sharper than \\(p(s | o_d)\\).
>
> - 2. **Enhanced Posterior Constraint**:
>       With a more concentrated posterior \\(p(s | o_d, o_r)\\), the KL divergence between the predicted distribution \\(q(s)\\) and the posterior is reduced:
>       \\[
>       D_{KL}(q(s) \| p(s | o_d, o_r)) < D_{KL}(q(s) \| p(s | o_d)).
>       \\]
>
>       This reduction in KL divergence reflects the enhanced constraint provided by the HQ reference, which reduces uncertainty and minimizes free energy.
>
> - 3. **Comparison of Free Energy**:
>    - To sum up, adding the HQ reference image reduces posterior uncertainty, leading to:
>    \\[
>    F_{\text{with ref}} = D_{KL}(q(s) \| p(s | o_d, o_r)) < F_{\text{no ref}} = D_{KL}(q(s) \| p(s | o_d)).
>    \\]
> ---
> **4. Conclusion**
>
> - In summary, based on the Free Energy Principle, utilizing feature differences between HQ and LQ images reduces posterior distribution uncertainty, minimizes free energy, and enhances perceptual prediction accuracy, thereby justifying the computation of these differences.
> ---
> References
>
> [1] The free-energy principle: a unified brain theory? Nature 2010
>
> [2] Using free energy principle for blind image quality assessment TMM 2014

---

> ### Author Response · Authors · 2024-11-23
> **Author Responses (3/n)**
>
> > **W4 (2)**: No ablation study verifies the effectiveness of the reference image.
>
> (2) To further demonstrate this, the following table presents the performance of two NR-IQA methods extended to the NAR-IQA setting (NAR-IQA needs to calculate feature differences between HQ and LQ images). The consistent improvements across all datasets highlight the effectiveness of using feature differences.
>
> |Model|LIVE|CSIQ|TID2013|KonIQ-10K|
> |-|-|-|-|-|
> | WaDIQaM-NR    | 0.855/0.855/0.656 | 0.716/0.750/0.527 | 0.585/0.610/0.416 | 0.382/0.386/0.261 |
> | WaDIQaM-NAR   | **0.897/0.894/0.707** | **0.799/0.851/0.613** | **0.670/0.694/0.493** | **0.362/0.364/0.258** |
> | DEIQT-NR      | 0.901/0.879/0.724 | 0.795/0.805/0.606 | 0.688/0.673/0.495 | 0.503/0.491/0.351 |
> | DEIQT-NAR     | **0.915/0.889/0.735** | **0.810/0.810/0.611**  | **0.733/0.728/0.547** | **0.508/0.498/0.353**|
>
> > **Q1**: How do the authors conduct model selection and hyperparameter tuning?
>
> To clarify, in IQA, due to the limited number of training data, most SOTA methods (e.g., HyperNet, DEIQT, LoDa) do not use a validation set. For fairness, we align our model selection and hyperparameter tuning with DEIQT, using the same learning rate across all datasets. Additionally, averaging performance over 10 runs further ensures robustness and fairness.
>
> > **Q2**: Generalization should be evaluated by training on authentic distortions and testing on synthetic ones, or vice versa.
>
> Thank you for your valuable suggestion. The results summarized in the table below show that our proposed method performs consistently well when training and testing on datasets with different distortion types. Notably, it achieves a maximum improvement of 14.61%, demonstrating strong generalization capabilities.
>
> | Training | Testing | DEIQT (PLCC/SRCC) | RKIQT (PLCC/SRCC) |
> |----------|---------|-------------------|-------------------|
> | KADID    | LIVEC   | 0.465 / 0.425    | **0.523 / 0.494**    |
> | KADID    | KONIQ   | 0.507 / 0.519    | **0.581 / 0.566**    |
> | KONIQ    | KADID   | 0.534 / 0.511    | **0.564 / 0.540**    |
> | KONIQ    | TID2013 |  0.531 / 0.466     | **0.550 / 0.473**    |
>
>
> > **Q3**: Why is there a discrepancy in SRCC performance on TID2013 between Table 1 and Table 8?
>
> Apologies for any confusion caused. The correct SRCC value for TID2013 is 0.900, and we will updated it in the revised version.
>
> > **Q4**: In Line 478, the reported result of 0.86 falls outside the range of 0.32 to 0.7. Please clarify.
>
> We apologize for any confusion caused. We want to emphasize that the similarity between the CLS token and the CNN token, as well as between the CLS token and the INN token, is relatively small, with values of 0.36, 0.32, 0.67, and 0.55, respectively. These results underscore the complementary nature of the CNN and INN tokens relative to the CLS token. To avoid ambiguity, we will revise this section in the updated manuscript for greater clarity.
>
> > **Q5**: In Fig. 4's second-to-last row, activation maps miss distortions.
>
> According to the perceptual rules of the human visual system (HVS), the HVS demonstrates a higher tolerance for distortions in relatively uniform backgrounds [1]. Instead, it tends to focus more on distortions in salient regions, such as foreground objects, where attention is naturally drawn [2, 3]. Consequently, in the second-to-last row of Figure 4, since the background is a flat wooded area, our method primarily focuses on sharpening the bird, which aligns with this perceptual tendency.
>
> ---
> References
>
> [1] Blindly Assess Image Quality in the Wild Guided by A Self-Adaptive Hyper Network. CVPR 2020
>
> [2] Saliency-Guided Transformer Network combined with Local Embedding for No-Reference Image Quality Assessment. ICCV 2021
>
> [3] Vsi: A visual saliency-induced index for perceptual image quality assessment. TIP 2014

---

> ### Author Response · Authors · 2024-11-26
> **Deadline reminder and looking forward to discussion**
>
> Dear Reviewer zofg,
>
> We deeply appreciate your efforts in reviewing our paper, and we are particularly grateful for the invaluable suggestions that have greatly enhanced its quality.
>
> To address your concerns, we have explained the role of inductive biases in preventing overfitting **(W1)** and clarified inconsistencies in statements and results **(W2)**. Additionally, we have discussed how our KD strategy differs from CVRKD **(W3)** and supplemented theoretical and experimental evidence for using non-aligned reference images **(W4)**. We also clarified our approach to model selection and hyperparameter tuning **(Q1)**, explored cross-dataset validation experiments between synthetic and real datasets **(Q2)**, addressed numerical discrepancies in charts **(Q3, Q4)**, and analyzed feature visualizations **(Q5)**.
>
> We kindly request that you review our rebuttal and let us know if further clarification is needed. Your insights and suggestions are crucial for improving our work. Thank you for your time and dedication!
>
> Best regards,
>
> Authors

---

> > ### Comment · Reviewer_zofg · 2024-11-26
> >
> > Thank you for your response. The technical contribution of this paper is not novel, which aligns with the comments made by other reviewers. Furthermore, the free energy principle is irrelevant to this work, as the HQ reference image differs significantly from the LQ distorted image, rendering the subtraction meaningless in the feature domain. The performance of the cross-dataset setting is relatively low, indicating the potential over-fitting issue of the model.  Additionally, lots of the given activation maps do not reflect the distortions accurately.

---

### Official Review · Reviewer_a6Ex · 2024-11-02

**Soundness:** 2
**Presentation:** 2
**Contribution:** 3
**Rating:** 3
**Confidence:** 4

**Summary:**

This work aims to evaluate the perceptual quality of images, and proposes a distillation method from non-aligned reference. The work introduces a teacher module with non-aligned reference, an inductive bias regularization, and a masked quality-contrastive distillation. The experiments show a good performance and the method almost outperforms existing methods.

**Strengths:**

1. The distillation from multi-networks and reference-based methods is good for blind image quality assessment.
2. Extensive experiments has been conducted to demonstrate the effectiveness.

**Weaknesses:**

1. The writing is not easy to understand, and the contributions are decentered. It is hard to connect the content within a core idea, which makes the manuscript lack of the main motivation.
2. Fig.2 seems disorder, and it would be good the reshape the work.
3. Some experimental results are missing. For example, Re-IQA in Tab 1 lacks the results on LIVEFB. I suppose the work has published its performance on the database. And if the method is reimplemented by the authors, it would be weird to miss it.
4. Also, the work seems too complicated, but still cannot perform superior over some simple models on several large-scale IQA datasets (e.g., KADID, LIVEFB). And it would be noted that the performance on KonIQ and SPAQ is merely similar to SAMA (2024-aaai), which is simply a sampling strategy.

**Questions:**

1. The authors may give a concise description on the main contribution of the work.
2. It is weird that some results are missing. For example, Re-IQA has published the original result on LIVEFB in the original paper, and the source code is also available, but the performance is not shown in Tab.1.

---

> ### Author Response · Authors · 2024-11-23
> **Author Responses (1/n)**
>
> > **W1, Q1**: A brief description of the motivation and main contributions.
>
> (1) Our core motivation are as follows:
> - While reference-based IQA methods are effective, they rely on high-quality reference images that are often unavailable in real-world scenarios. Existing NR-IQA methods, however, perform suboptimally due to insufficient use of comparative knowledge. Therefore, we aim to transfer the comparative knowledge from reference-based IQA to NR-IQA, achieving strong performance without relying on reference images.
>
> (2) Our main contributions are as follows:
> - We found that **introducing comparative awareness into NR-IQA is crucial**, as it aligns more closely with human perception and has the potential to provide a general framework orthogonal to existing NR-IQA architectures.
>
> - We propose MCD to capture differential prior information between HQ and LQ images, enhancing its ability to **perceive quality differences of LQ images relative to HQ images.** MCD trains different local pixel regions to perceive quality differences by masking certain pixels. This not only mitigates the negative impact of traditional MSE loss caused by the inconsistency between HQ-LQ features and LQ features but also significantly enhances the quality perception of local pixels.
>
> - We propose an inductive bias regularization method to **reduce overfitting (e.g., mitigating the impact of changes in reference image content in MCD) during training** by efficiently capturing both local distortions and global features of the distorted image. To support this process, we further introduce **learnable intermediate layers for reverse distillation**, which help bridge the gap between teachers and students with different inductive biases, thereby enhancing the student's ability to effectively learn both local and global quality attributes.
>
> - Experiments on standard IQA datasets demonstrate that our RKIQT achieves superior performance compared to IQA methods that rely on reference images, such as CVRKD and IQT-NAR.
>
>
> > **W2**: Fig.2 seems disorder, and it would be good the reshape the work.
>
> Thank you for your insightful suggestion. We will modify Figure 2 in the revised version.
>
> > **W3,Q2**: The experimental results of Re-IQA on LIVEFB are missing.
>
> Valuable feedback! In the original Re-IQA paper, the performance was reported under a specific data split, whereas we compare the average performance across 10 random splits. To ensure fairness, we initially did not include its results in Table 1. Following your suggestion, we have reproduced the results of Re-IQA on the LIVEFB dataset under the same 10 random splits. As shown in the table below, our method still demonstrates a clear advantage.
>
> | Method      | LIVEFB (PLCC)  | LIVEFB (SRCC) |
> |-|-|-|
> | Re-IQA      | 0.667  | 0.576  |
> | RKIQT       | **0.686**  | **0.589**  |
>
> > **W4**: The method seems overly complex but doesn't outperform simpler models on large-scale IQA datasets (e.g., KADID, LIVEFB) and performs only comparably to SAMA (2024-AAAI) on KonIQ and SPAQ.
>
> We appreciate the reviewer’s feedback on the complexity and performance of our method. Below, we address these points:
>
>   - **1. Complexity of the Proposed Method:** To clarify, the complexity of our method primarily lies in the cost of pretraining the teacher model, while the actual deployment is straightforward and adds only about 4M extra parameters. To further illustrate, we provide an overview of RKIQT's computational costs. The proposed RKIQT method is not overly complex; in fact, it achieves higher throughput compared to SAMA and LoDa, demonstrating faster inference speeds.
>
> |Method | Encoder|Total Params (M)|MACs (G)|Throughput (pairs/s)|
> |:------:|:-------:|:------:|:------:|:----------:|
> |LoDa| ViT-B  | 118 | 23.0 | 258 |
> |SAMA| Swin-T  | 28 | 5.9  | 114 |
> |RKIQT  |ViT-S | 28 | 27.3  | 301 |
>
>   - **2. Performance on KADID and LIVEFB:** In IQA, even small improvements matter. DEIQT improved SRCC on LIVEFB by **0.5 points**, while our method achieved **1.1 points**. On KADID, LoDa and LIQE outperform RKIQT but use 118M and 151M parameters compared to RKIQT's 28M. Scaling RKIQT to ViT-B (Table 9) boosts SRCC by **0.9 points**, surpassing LoDa.
>
> | Method| CSIQ| KADID | LIVEC|LIVEFB|
> |-|-|-|-|-|
> | LoDa (118M)  | - / - |  0.920 / 0.912  | 0.899 / 0.876 | 0.679 / 0.578 |
> | SAMA (28M)  |  0.962 / 0.960 |  0.905 / 0.903  | 0.881 / 0.847 | 0.662 / 0.576 |
> | RKIQT (28M)  | 0.970 / 0.958 | 0.911 / 0.911 | 0.917 / 0.897 | 0.686 / 0.589 |
>
>   - **3. Comparison with SAMA:**
>   - Although SAMA is only a simple sampling method, it may bring some additional inference burden due to additional interpolation and mask on the data during the actual deployment. As shown in the table above, SAMA has slower inference speed.
>   - Generalization: As shown in the table above, RKIQT performs robustly on different datasets, including LIVEC,CSIQ, LIVEFB, and KADID.

---

> ### Author Response · Authors · 2024-11-26
> **Deadline reminder and looking forward to discussion**
>
> Dear Reviewer a6Ex,
>
> We deeply appreciate your efforts in reviewing our paper, and we are particularly grateful for the invaluable suggestions that have greatly enhanced its quality.
>
> In response, we have emphasized our key motivations and contributions **(W1, Q1)**, supplemented additional experimental results on the LIVEFB dataset for Re-IQA **(W3, Q2)**, and analyzed the complexity and advantages of our method compared to SAMA **(W4)**.
>
> We kindly request that you review our rebuttal and let us know if further clarification is needed. Your insights and suggestions are crucial for improving our work. Thank you for your time and dedication!
>
> Best regards,
>
> Authors

---

### Official Review · Reviewer_jj1w · 2024-11-02

**Soundness:** 3
**Presentation:** 3
**Contribution:** 2
**Rating:** 5
**Confidence:** 5

**Summary:**

This paper seeks to solve the no-reference image quality assessment issue with the help of distilled reference knowledge while eliminating the direct use of reference images. As such, they propose the Reference Knowledge-Guided Image Quality Transformer scheme that guides a student model to emulate the teacher's prior knowledge. Besides, to mitigate the weakness in local structures and inductive biases of ViTs, they further propose a Masked Quality-Contrastive Distillation method, benefiting from both CNNs and INNs.

**Strengths:**

The proposed model achieves some promising results compared to NR IQA methods, even with some FR peers.
The complementary strategy between ViTs and CNN/INNs is creative.

**Weaknesses:**

1. The idea of utilizing pseudo reference images, including both image-level or feature-level and both content-relevant or -irrelevant settings, in no-reference image quality assessment methods has been a longstanding topic all the time. The mentioned flaws of the listed relevant works Liang 2016 and Yin 2022 (the third paragraph of the Introduction), i.e., requiring suitable high-quality images as references during quality inference, have been widely noticed and addressed in the related works such as:
[1] Chen B, Zhu L, Kong C, et al. No-reference image quality assessment by hallucinating pristine features[J]. IEEE Transactions on Image Processing, 2022, 31: 6139-6151.
[2] Tian Y, Chen B, Wang S, et al. Towards Thousands to One Reference: Can We Trust the Reference Image for Quality Assessment?[J]. IEEE Transactions on Multimedia, 2023.
[3] Ma J, Wu J, Li L, et al. Blind image quality assessment with active inference[J]. IEEE Transactions on Image Processing, 2021, 30: 3650-3663.
However, they are not reviewed or mentioned in this paper. Therefore, the novelty of this work is my major concern. To justify this, providing comparisons with similar models, both theoretically and practically, would increase the demonstration of novelty.
2. The current experiments cannot fully support the claimed contributions , requiring further refinement.

**Questions:**

1. There are missing comparisons with current SOTAs. Would you provide the performance comparison with TOPIQ and FPR (indicated as follows) on the involved datasets?
[1] Chen C, Mo J, Hou J, et al. Topiq: A top-down approach from semantics to distortions for image quality assessment[J]. IEEE Transactions on Image Processing, 2024.
[2] Chen B, Zhu L, Kong C, et al. No-reference image quality assessment by hallucinating pristine features[J]. IEEE Transactions on Image Processing, 2022, 31: 6139-6151.
2. In the non-aligned reference teacher model, the reference information is learned by first computing the difference between content-irrelevant LQ and HQ images. Please provide more information on this process. How are the pairs formulated? How many HQ images are needed for each LQ image? What is the influence of formulating such LQ-HQ pairs differently?
3. Further, for the non-aligned reference teacher model, would the authors provide the experimental results indicating the insensitivity of the reference images incorporated for training?
4. Why do you incorporate the three different inductive bias tokens? Are they equally important?
5. The NAR-teacher network is trained on the KADID dataset and is employed throughout all the experiments. Meanwhile, the proposed model achieves promising results on all the artificially distorted image datasets while showing sub-optimal performance on authentic images. Is this because the NAR module implicitly learns the distortion information while pretraining? If the reference images are altered to the highest-quality images in authentic datasets, will the proposed method still work?

**Details Of Ethics Concerns:**

Not applicable.

---

> ### Author Response · Authors · 2024-11-23
> **Author Responses (1/n)**
>
> > **Weaknesses 1:** Comparison with [1] FPR, [2] FLRE, and [3] AIGQA methods.
>
> 1. Firstly, let us outline the similarities between our proposed approach RKIQT and [1,2,3]:
>    - Both RKIQT and [1, 3] fall under the category of NR-IQA, as they do not rely on reference images during inference. Conversely, [2] employs a Full-Reference IQA framework, necessitating pristine reference images during inference.
>    - Inspired by HVS, RKIQT and [1,2,3] proposed the IQA method for learning reference knowledge.
>
> 2. Next, let us outline the differences between our proposed approach RKIQT and [1,2,3]:
>   - ##### a. Comparative Learning rather than Reconstruction
>     - **[3]:** Utilizes a generative network to reconstruct pristine reference images from distorted inputs.
>     - **[1]:** Learning generates pseudo-pristine reference features directly from distorted image features.
>     - **Limitations of [1, 3]:** Both methods rely on directly inferring or reconstructing the pristine reference image's knowledge from distortion information, which becomes increasingly challenging as the severity and complexity of distortions vary in real-world scenarios.
>     - One key difference between the proposed method and [1] and [3] is **its focus on comparative learning rather than direct reconstruction.**  Specifically, the method trains the model to capture differential prior information between HQ and LQ images, enhancing its ability to perceive quality differences of LQ images relative to HQ images, instead of emphasizing the reconstruction of pesudo-reference image information. This approach not only reduces the complexity of training but also aligns more closely with human perceptual mechanisms. As a result, the model achieves more robust and accurate quality predictions, even under complex distortion scenarios.
>
>   - ##### b. The degree of demand for the reference image
>     - Both [1,2,3] methods rely on the original reference image for supervision, requiring the model to explicitly learn to reconstruct the original reference image or its features. However, when the original reference image is unavailable in real-world scenarios, methods [1,2,3] face limitations.
>     - To address this issue, we leverage content-unaligned reference images to help RKIQT focus on learning HQ-LQ differential features rather than reconstructing the pristine reference image. This allows our method to utilize a broader variety of high-quality images as reference knowledge, making it more adaptable to real-world applications.
>
>   - ##### c. Regularization to Prevent Overfitting
>     - We propose an inductive bias regularization method to reduce overfitting (e.g., mitigating the impact of changes in reference image content in MCD) during training by efficiently capturing both local distortions and global features of the distorted image. Additionally, a learnable intermediate layer bridges the inductive bias gap between teacher and student models, enabling more effective knowledge transfer.
> 3. We reiterate the main contributions of our work:
>   - We found that **introducing comparative awareness into NR-IQA is crucial**, as it aligns more closely with human perception and has the potential to provide a general framework orthogonal to existing NR-IQA architectures.
>   - **Masked Quality-Contrastive Distillation (MCD):** This method enhances local quality sensitivity by training on masked pixel regions, reducing the negative effects of traditional MSE losses due to inconsistencies between high-quality (HQ) and low-quality (LQ) features.
>   - **Inductive Bias Regularization:** By leveraging networks with varying inductive biases as teacher models and employing reverse distillation through a learnable intermediate layer, our method effectively and adaptively adjusts the inductive bias of the student model, facilitating the learning of both local and global features.
>   - **Extensive Validation:** Experiments on benchmark IQA datasets confirm that RKIQT surpasses traditional IQA methods that require reference images (e.g., CVRKD and IQT-NAR), demonstrating its effectiveness and practicality.
>
> These modules collectively contribute to the superior performance of our RKIQT across multiple datasets compared to methods such as FPR [1], FLRE [2], and AIGQA [3], as shown in the table below.
> | Method      | LIVE         | CSIQ         | TID2013      | KADID        |
> |-------------|--------------|--------------|--------------|--------------|
> | AIGQA [3]   | 0.957/0.960  | 0.952/0.927  | 0.893/0.871  | -            |
> | FPR [1]     | 0.974/0.969  | 0.958/0.950  | 0.882/0.854  | 0.898/0.894  |
> | FLRE [2]    | 0.970/0.964  | 0.960/**0.966**  | 0.899/0.876  | -            |
> | RKIQT       | **0.986/0.984**  | **0.970**/0.958  | **0.917/0.897**  | **0.911/0.911**  |
>
>
> > **Weaknesses 2:**: The current experiments cannot fully support the claimed contributions.
>
> Please kindly refer to the response to Q1~Q5.

---

> ### Author Response · Authors · 2024-11-23
> **Author Responses (2/n)**
>
> > **Question 1:**: There are missing comparisons with TOPIQ and FPR.
>
> The experimental results are presented in the table below. Notably, since TOPIQ is designed for the FR-IQA setting, we only compared its performance under the NR-IQA setting to ensure fairness. Our method demonstrates superior performance across multiple datasets.
> | Method      | LIVEC        | KonIQ        |
> |-------------|--------------|--------------|
> | TOPIQ       | 0.884/0.870  | 0.939/0.926  |
> | RKIQT       | **0.917/0.897**  | **0.943/0.929**  |
>
> | Method      | LIVE         | CSIQ         | TID2013      | KADID        |
> |-------------|--------------|--------------|--------------|--------------|
> | FPR     | 0.974/0.969  | 0.958/0.950  | 0.882/0.854  | 0.898/0.894  |
> | RKIQT       | **0.986/0.984**  | **0.970/0.958**  | **0.917/0.897**  | **0.911/0.911**  |
>
> > **Question 2:**: (1) How does the non-aligned reference teacher model compute reference information from content-irrelevant LQ-HQ differences?
> (2) How are the LQ-HQ pairs formulated?
> (3) How many HQ images are required per LQ image?
> (4) What is the impact of using different LQ-HQ pairing strategies?
>
> (1). **Reference Information Calculation Process**: Given a low-quality (LQ) input image \\(I_L\\) and a high-quality (HQ) reference image \\(I_H\\), feature maps \\(F_L\\) and \\(F_H\\) are first extracted using the Inception-ResNet-v2 model. The difference between the feature maps is then computed as \\(F_{\text{diff}} = F_H - F_L\\).
>
> (2). **Formulation of LQ-HQ Pairs**: To provide high-quality image, we use the DIV2K HR dataset, which contains 1,000 high-resolution images. For each LQ image, an HQ image is randomly sampled from the DIV2K HR dataset to create the corresponding LQ-HQ pair.
>
> (3). **Number of HQ Images per LQ Image**: Each LQ image requires only one HQ image as a reference.
>
> (4). **Impact of LQ-HQ Pair Formulations**: As shown in Table 8 of the manuscript, we investigate the impact of three different LQ-HQ pair configurations on the performance of the student model. Specifically:
>    - **Content-align:** The HQ image is the original clear version of the LQ image, ensuring perfect content alignment.
>    - **Content-similar:** The HQ image is generated by applying affine transformations to the content-aligned image, including random scaling factors (\\(s\\)) within [0.95, 1.05] and rotations (\\(\theta\\)) within [\\(-5^\circ, 5^\circ\\)], resulting in partially misaligned content.
>
> |                   | TID    |       | CSIQ  |       |
> |-------------------|--------|-------|-------|-------|
> |                   | PLCC   | SRCC  | PLCC  | SRCC  |
> | Content-align     | 0.941  | 0.928 | 0.973 | 0.963 |
> | Content-similar   | 0.926  | 0.909 | 0.965 | 0.960 |
> | Content-dissimilar| 0.917  | 0.900 | 0.970 | 0.958 |
>
> The results demonstrate that using HQ images with higher content similarity improves model performance further. Notably, even when using content-different HQ images (which are easier to obtain in real-world scenarios), the RKIQT model still achieves state-of-the-art performance. This highlights the effectiveness of the proposed approach in learning from reference knowledge.
>
> > **Question 3:**: For the non-aligned reference teacher model, would the authors provide the experimental results indicating the insensitivity of the reference images incorporated for training?
>
> Thank you for your valuable feedback! Based on your suggestion, we conducted further experiments to study the sensitivity of the NAR-teacher network to reference images, as shown in the table below. Specifically, **we selected the top 10% highest-quality images (based on quality scores) and the distorted images from the SPAQ datasets** to form LQ-HQ pairs randomly. The teacher network was then trained exclusively on these LQ-HQ pairs. The results indicate that the performance of the teacher network remains relatively stable, demonstrating its robustness to variations in reference images.
>
> | Reference    | LIVE           | CSIQ           | TID2013        | KonIQ-10K     |
> |--------------|----------------|----------------|----------------|---------------|
> | w/ SPAQ HR   | 0.909 / 0.890 / 0.731 | 0.813 / 0.832 / 0.617 | 0.681 / 0.702 / 0.495 | 0.461 / 0.464 / 0.316 |
> | w/ DIV2K HR  | 0.903 / 0.888 / 0.717 | 0.799 / 0.821 / 0.609 | 0.674 / 0.691 / 0.490 | 0.470 / 0.472 / 0.322 |

---

> > ### Comment · Reviewer_jj1w · 2024-11-26
> >
> > Thanks for the detailed elaborations on the difference between the two compared with the mentioned paper.
> >
> > As reflected by the author's rebuttal, the core difference is whether the reference image participates in the quality evaluation by itself or by the reconstructed version. Instead, the proposed approach in this paper tends to capture the difference in quality with unmatched content images. This is very similar to the often-called unpaired learning.
> >
> > However,  as declared by the rebuttal, the LQ and HQ images are transformed into the feature space and are directly subtracted to obtain the residual. And the performance of switching to unpaired content does not drop a lot according to the provided Table. The reviewer is confused about this step since the quality information is highly relevant to image content, causing it to be tough to capture quality differences with such residuals. I can hardly find similar practices in other deep-learning-based low-level vision models. Moreover, the specific formulation process of the LQ-HQ image pairs is still absent.
> > Besides, the added experimental results show a surprisingly coherent tendency on different datasets, which is very interesting.
> >
> > The authors are highly recommended to:
> > - Reconsider the reasonability of the proposed pipeline;
> > - Provide theoretical proof or previous practice in the literature to show that quality information can be necessarily obtained by directly learning on the residuals of unpaired content.
> > - Add the missing but relevant literature;
> > - Re-verify the experimental results.
> >
> > The rebuttal does not settle my questions but brings in new concerns. Therefore, I cannot change my score on this paper.

---

> ### Author Response · Authors · 2024-11-23
> **Author Responses (3/n)**
>
> > **Question 4:**: (1) Why do you incorporate the three different inductive bias tokens? (2) Are they equally important?
>
> (1). We aim to ensure that the tokens explicitly carry distinct inductive biases, enabling them to learn more effectively from teachers with corresponding inductive biases.
> (2). These tokens are **equally important**, as they allow the student model to acquire diverse knowledge from teachers with different inductive biases, thereby improving both performance and generalization capabilities.
>
> > **Question 5:**: (1) Is the RKIQT sub-optimal performance on authentic images due to learning distortion information during pretraining?
> (2) Can the method perform well with the highest-quality images from authentic datasets as references?
>
> （1）Indeed, the RKIQT's performance on authentic datasets is influenced by the characteristics of the pre-trained synthetic dataset. Specifically, synthetic distortions are typically globally uniform, leading the NAR-teacher to prioritize global distortion patterns, such as those affecting edges or contours (Figure 4). However, authentic distortions are often more locally non-uniform, presenting a unique challenge for capturing fine-grained patterns.  Additionally, it is important to clarify that even small improvements are significant in real-world IQA datasets. DEIQT improves SRCC by 0.5 points on LIVEFB and 0.5 points on KonIQ, whereas our method achieves improvements of **1.1 points** and **0.8 points**, respectively.
>
> （2）Using the highest-quality images from authentic datasets as reference images is still effective. Based on your suggestion, we conducted additional experiments by forming LQ-HQ pairs using the top 10% of highest-quality images **(IQA-HQ)** from the KonIQ, LIVEC, and SPAQ datasets. The student network was then trained exclusively on these LQ-HQ pairs.
> The results show that different reference images remain effective, and the RKIQT model is robust to the source of high-quality images.
> ||KonIQ||LIVEC||SPAQ||
> |-|-|-|-|-|-|-|
> |Method| PLCC|SRCC|PLCC|SRCC|PLCC|SRCC|
> |w/ IQA-HQ |0.940|0.927|0.915|0.895|0.929|0.924|
> |RKIQT|0.943|0.929|0.917|0.897|0.928|0.923|

---

> ### Author Response · Authors · 2024-11-26
> **Deadline reminder and looking forward to discussion**
>
> Dear Reviewer jj1w,
>
> We deeply appreciate your efforts in reviewing our paper, and we are particularly grateful for the invaluable suggestions that have greatly enhanced its quality.
>
> In response to your comments, we have clarified the key distinctions between our method and other IQA approaches **(Q1)** and included performance comparisons with related works **(W1, Q1)**. Additionally, we have provided detailed explanations regarding the selection of HQ-LQ image pairs **(Q2)** and conducted ablation studies to assess sensitivity to reference images **(Q3)**. Furthermore, we have explained the necessity of inductive bias tokens **(Q4)** and explored the effectiveness of using high-quality reference images from real-world datasets **(Q5)**.
>
> We kindly request that you review our rebuttal and let us know if further clarification is needed. Your insights and suggestions are crucial for improving our work. Thank you for your time and dedication!
>
> Best regards,
>
> Authors

---

### Meta-Review · Area_Chair_ZdYV · 2024-12-18

**Metareview:**

This paper proposes a RKIQT by introducing the reference knowledge into the NR-IQA. Experimental results show the effectiveness of the proposed method. The major concerns of the reviewers include the  limited novelty,  insufficient evaluations and comparisons with state-of-the-art methods.

The provided rebuttal does not solve the concerns of reviewers. Based on the recommendations of reviewers, the paper is not recommended acceptance.

**Additional Comments On Reviewer Discussion:**

During the discussions, the major concern about the limited novelty still remains. In addition,  Reviewer Qsu4 still concerns whether the comparisons are fair or not. The proposed method exists the potential over-fitting issue as pointed out by Reviewer zofg.

As these concerns are critical, the authors do not solve them well.

---

### Decision · Program_Chairs · 2025-01-22

Reject